# SIMILARITY AND SEPARATION OF LAST-ITERATE CONVERGENCE BETWEEN OPTIMISM AND REFLECTED ALGORITHMS IN TIME-VARYING GAMES

## ABSTRACT

In this paper, we investigate the behaviours of reflected gradient (RG), accelerated reflected gradient (ARG), and optimistic gradient (OG) algorithms in multi-player games modelled as variational inequalities with $L$-smooth continuous monotone limits in convergent time-varying cases and with $L$-smooth continuous and monotone games at each time in periodic cases, both in convex action sets. The RG, ARG, and OG algorithms require fewer complex calculations, i.e., on the gradients and projections per iteration. We prove that a convergence rate of $O(1/\sqrt{T})$ and $O(1/T)$ can be reached by the RG and ARG algorithms with bounded action sets for convergent perturbed monotone games, respectively, if the sequence of time-varying games converges to the limit fast enough, without additional assumptions like strong monotonicity, and such a result matches and improves the existing results on similar algorithms requiring calculations on two gradients with different actions. Besides, a surprising result is also shown that the standard OG algorithm in time-varying games behaves dramatically differently from its variant and other similar algorithms: the standard OG algorithm converges in any sequences of time-varying monotone $L$-smooth games with a common Nash equilibrium, including some periodic games, while at the same time, its variant with a slight difference diverges exponentially even in periodic games. We also show that the RG and ARG algorithms diverge exponentially in some periodic games.

## 1 INTRODUCTION

This paper discusses machine learning algorithms applied to a type of repeated multi-player game. In this type of game, each player selects an action from a continuous closed convex action set, and pays a certain cost depending on all players' actions. The relationship between these actions and costs is determined by any one game of this type evolving over time and is unknown to any player a priori. As the game evolves, each player changes their action according to their cost and other known information, followed by the game repeating.

We seek to address the following open questions: Do improved versions of the extra gradient (EG) algorithm with only one calculation on the gradient or the projection, such as the reflected gradient (RG) algorithm, the accelerated reflected gradient (ARG) algorithm and the optimistic gradient (OG) algorithm, converge in time-varying monotone games with constant step sizes? If so, how fast do these algorithms converge?

### 1.1 BACKGROUND

Machine learning and game theory problems have intrinsic connections (Zhu et al. (2023); Lv et al. (2023)). Monotone games, a class of multi-player games introduced by (Rosen (1965)), include extensively discussed games such as two-player zero-sum games, convex-concave games (both including two-player bilinear games (Feng et al. (2023; 2024)), $\lambda$-cocoercive games (Lin et al. (2020)), zero-sum polymatrix games (Daskalakis & Papadimitriou (2009); Cai & Daskalakis (2011)), and zero-sum socially-concave games (Even-dar et al. (2009)).

The problem of finding a Nash equilibrium in a smooth monotone game is equivalent to solving a Lipschitz and monotone variational inequality (MVI). Many gradient-based algorithms have been proposed to solve game problems modelled as monotone variational inequality problems. A well-known result on the optimal convergence rate of such algorithms in general MVI problems is $O(1/T)$ achieved by the EG algorithm (Korpelevich (1976)) and its Bregman variant, the Mirror-Prox (MP) algorithm (Nemirovski (2004)).

A major disadvantage of the EG algorithm is the requirement of two calculations on projections and gradients per iteration, which means it requires more computation than the gradient descent (GD) algorithm due to the significant computational cost on gradients and on projections in constrained games. Hence, its improvements, avoiding calculating the gradient and projection a second time like in the EG algorithm while retaining the algorithm's "predictive" properties, have attracted much interest from the machine learning community. There are two complementary approaches to reduce the costly calculations on reducing the number of calculations of projections and gradients per iteration. The OG algorithm (Tseng (2000)) applies the former approach, and the PEG algorithm (Popov (1980)) focusing on gradient extrapolation mechanisms applies the latter. The RG (Malitsky (2015)) and the newly proposed ARG (Cai & Zheng (2023)) algorithms combine both.

Unfortunately, in time-varying game problems, it is still inevitable to calculate each of the gradient and the projection twice per iteration with the past extra gradient (PEG) algorithm (also known as the optimistic gradient descent ascent (OGDA) algorithm in some papers Feng et al. (2023)), which increases the computational cost and makes the PEG algorithm as inefficient as the EG one. In contrast, both the RG algorithm and the ARG algorithm developed from it require calculating each of the gradient and the projection only once per iteration, and the OG algorithm requires calculating gradients twice, but the projection only once, per iteration. Inspired by the advantages of the RG, ARG, and OG algorithms, we investigate their convergence behaviors to analyze whether the three algorithms perform satisfactorily for time-varying games, a complex problem.

## 1.2 RELATED WORKS

### 1.2.1 PREVIOUS RESEARCH ON THE EXTRA GRADIENT ALGORITHM AND ITS VARIANTS

Most known machine learning algorithms applied to game problems are based on gradients. The simplest one of them is the GD algorithm. However, the GD algorithm diverges even for time-invariant two-player zero-sum bilinear games (Daskalakis et al. (2018)). To eliminate this limitation, many improved learning algorithms have been proposed, including the EG algorithm (Korpelevich (1976); Cai et al. (2022); Feng et al. (2023); Monteiro & Svaiter (2010)) and its variants like the PEG (Popov (1980); Cai et al. (2022); Feng et al. (2023)), OG (Daskalakis et al. (2018)), and RG (Chambolle & Pock (2011); d'Angelo et al. (2014); Hsieh et al. (2019)) algorithms. Other algorithms like the negative momentum algorithm (Feng et al. (2023)) and weights update algorithms (Arora et al. (2012); Cai et al. (2024a)) are also applied by some researchers.

For the two classical algorithms, the EG (Korpelevich (1976)) and PEG (Popov (1980)) algorithms, their convergence rates in time-invariant games have been thoroughly investigated. The convergence property of the EG algorithm has been investigated in both concave games (Monteiro & Svaiter (2010) on last-iterate convergence and Nemirovski (2004)) and special non-concave games (Mertikopoulos et al. (2019)). The EG, PEG and other related algorithms in unconstrained strongly monotone games and unconstrained bilinear games have also been proven to have linear convergence rates (Daskalakis et al. (2018)). Later papers have proved their asymptotic convergence (Daskalakis & Panageas (2019)) and last-iterate convergence for games with monotone gradients of the cost functions with the rate $O(1/\sqrt{T})$ (Golowich et al. (2020); Cai et al. (2022)). A recent result (Wei et al. (2021)) shows that there exists even always an exponential convergence rate for the PEG algorithm in time-invariant games. However, a surprising result, the separation of last-iterate convergence behaviors between the EG and PEG algorithms in time-varying games (Feng et al. (2023)), has also been shown recently.

Compared with the EG and PEG algorithms, investigations of the RG (Malitsky (2015)), OG (Tseng (2000)) and ARG (Cai & Zheng (2023)) algorithms proposed more recently are less extensive. (Yang & Liu (2018)) shows weak convergence of the RG algorithm with general closed convex action sets in monotone $L$-smooth games and its linear convergence with general closed convex action sets in strongly monotone $L$-smooth games. The RG algorithm has also been applied in some works and its performances have been shown to be satisfactory (Hsieh et al. (2019)). Cai & Zheng (2023) has

showed convergence rate of the RG algorithm to be $O(1/\sqrt{T})$ in time-invariant monotone games. The OG algorithm has been shown to strongly converge (Yang & Liu (2019)) in time-invariant monotone games, and Cai & Zheng (2023) showed that its best-iterate convergence rate is $O(1/\sqrt{T})$ with the same conditions. Dung et al. (2024) has proved the weak convergence of the OG algorithm with conditions weaker than monotone games. The ARG algorithm has been proposed very recently (Cai & Zheng (2023)) and its last-iterate convergence rate in time-invariant monotone games is proven to be $O(1/T)$ together. So far, investigations of that algorithm are rare. However, the thought applied to this algorithm, the Halpern iteration, is classical (Halpern (1967)). Such a technique is closely related to Nesterov's accelerated method (Tran-Dinh (2022)) and related methods have been extensively researched. Other literature on similar algorithms has been summarized in (Cai et al. (2024b)).

### 1.2.2 Previous research on time-varying games

In recent years, works on time-varying games have been emerging. Most of them focused on time-average convergence behaviors (Cardoso et al. (2019); Zhang et al. (2022); Yan et al. (2023)). To divert from the difficulties in calculating the Nash equilibrium (Daskalakis et al. (2009); Chen et al. (2009); Deligkas et al. (2022)), some researchers turned to a weakened concept, correlated equilibrium (Anagnostides et al. (2023)), with limited usage within the machine learning community. An important recent article by Duvocelle et al (Duvocelle et al. (2023)) has discussed strictly monotone games with decreasing step sizes and has shown that the probability of convergence is 1, despite not providing a convergence rate, a key measure of a learning algorithm. Decreasing step sizes is also unnatural since such a requirement considers new information decreasingly important instead of equally or increasingly as expected (Lin et al. (2020)). Hadiji et al. (2024) investigated strongly monotone games with contractive algorithms thoroughly, showing that the total errors though time (called "tracking errors") of contractive algorithms are sublinear if their optimal solution has sublinear quadratic path lengths and are logarithmic if they are periodic games with unique solutions with time-invariant step sizes and the online gradient descent algorithm.

However, the above results are based on the strong or strict monotonicity of variational inequalities, while simple games such as two-player zero-sum bilinear games are not even strictly monotone, with $\langle F(z_1) - F(z_2), z_1 - z_2 \rangle = 0$. For problems involving non-strictly monotone games, Feng et al. (2023) discussed unconstrained two-player zero-sum bilinear games without requiring decreasing step sizes and obtained the first known result on the last-iterate convergence in time-varying games. More recently, Feng et al. (2024) discussed constrained two-player zero-sum bilinear periodic games, and Chen & Yu (2025) has provided an important result that the EG and PEG algorithms are robust to convergent perturbation under the BAP assumption (defined in Assumption 1, Section 2.2, which means the time-varying game converges fast enough) for general convergent perturbed monotone games. Other known results involving games which are not necessarily strictly monotone include Zhang et al. (2022) focused on regret bounds (two-player) in time-varying bilinear saddle-point problems parameterized by the similarity of the payoff matrices and the equilibria of these games, Cardoso et al. (2019) providing an optimal solution based on the Nash equilibrium regret, and Duvocelle et al. (2023) discussing a general algorithm based on mirror descent for non-converging games and proved the convergence rate $O\left(T^{-(1-r)/3}\right)$ where $r < 1$.

Besides, for the algorithms requiring less computation per iteration like the OG, RG and ARG algorithms, no known literature exists on whether they converge in the average sense or exhibit last-iterate convergence in time-varying games, leaving a big gap.

### 1.3 Our contributions

Even though time-varying games are more realistic than time-invariant ones, existing results on convergence of machine learning algorithms in time-varying non-strong monotone games are limited. To fill such gaps, we investigate the last-iterate behaviors of the RG, ARG and OG algorithms in time-varying games. Our main results on their behaviors are summarized as follows.

- With methods inspired by (Yang & Liu (2018); Cai et al. (2022); Cai & Zheng (2023); Feng et al. (2023)), we prove that a time-invariant step size $\eta$ is sufficient for last-iterate convergence at the rates of $O(1/\sqrt{T})$ for the RG algorithm and $O(1/T)$ for the ARG algorithm in fast-converging perturbed time-varying monotone games under well-established

measures of performance, like tangent residuals, which expands the results in (Cai & Zheng (2023)). Note that the convergence rate $O(1/T)$ for the ARG algorithm matches the optimal convergence rate for all first-order methods for monotone inclusion problems (Diakonikolas (2020); Yoon & Ryu (2021); Cai & Zheng (2023)), which include the problems in this paper.

- We show that both the RG and ARG algorithms may diverge in a periodic monotone game even with a common Nash equilibrium.

- With methods inspired by (Cai & Zheng (2023); Feng et al. (2023)), we show that the OG algorithm and one of its variants behave surprisingly differently. Weak convergence of the standard OG algorithm is robust to time-varying games being monotone at each time as long as they share the same Nash equilibrium and they are all $L$-smooth. This is different from another optimistic algorithm called the OGDA (i.e. PEG) algorithm, which may diverge in periodic games (Feng et al. (2023)). However, the variant of the OG (VOG) algorithm with the first step of each iteration in the OG algorithm applying the gradient function used in the second step of the previous iteration may diverge even if the game is periodic.

A comparison of the RG, ARG, OG and VOG algorithms with the PEG and EG algorithms is shown in Table 1, where the results on convergent perturbed games are for games with bounded action sets for PEG, EG, RG and ARG algorithms and games with common Nash equilibria for PEG, EG, RG and OG algorithms, the results on periodic games are for games with common Nash equilibria for OG algorithms, the convergence rates of PEG, EG, RG and ARG algorithms are last-iterate convergence rates and the convergence rates of OG and VOG algorithms are best-iterate convergence rates.

**Remark 1.** *We discuss the periodic and convergent games in our paper for the following reasons. Both kinds of games have been discussed in previous literature as testing grounds for learning algorithms in the machine learning community (Duvocelle et al. (2023); Fiez et al. (2021); Feng et al. (2023; 2024); Chen & Yu (2025)), and last-iterate behaviours are well-defined in those kinds of time-varying games. Besides, our paper focuses on convergence behaviors of algorithms with less computational cost than those discussed in existing research, making it their natural extension. Moreover, the periodic game and the convergent perturbed game are natural generalizations of the time-invariant games in theoretical problems and in reality: a periodic game with the period of $1$ is a time-invariant game, and a convergent perturbed game with the perturbation of $0$ is also a time-invariant game. In theoretical problems, for the last iterate behaviours, the generalization of convergent perturbed games has been demonstrated in (Feng et al. (2023); Chen & Yu (2025)), and the generalization of periodic games has been demonstrated in (Feng et al. (2023)). In reality, the convergent perturbed game is a good model for the game affected by feedback progress, while for the periodic games, since seasonal changes or daily changes affect many ecological or market competitions in reality, such examples can be modelled as periodic games, and frameworks of multi-agent contextual games in (Sessa et al. (2020)) can also be modelled as periodic games.*

Our remaining paper is organized as follows. Section 2 defines the game problems, describes the main definitions and lemmas involved in these problems and introduces the model of machine learning in our game problems. Sections 3 and 4 show last-iterate convergence of the RG and ARG algorithms defined with tangent residuals in both periodic and BAP cases. Section 5 show best-iterate convergence of the standard OG algorithm and its variant. Section 6 concludes the paper with proposals on future directions.

## 2 PRELIMINARIES

In this section, we introduce the machine learning dynamics of time-varying games, which involves the RG, ARG and OG algorithms.

### 2.1 NOTATIONS

In this paper, the Euclidean space $(\mathbb{R}^n, \|\cdot\|)$ is considered, where $\|\cdot\|$ is the 2-norm. $\langle \cdot, \cdot \rangle$ denotes an inner product on $\mathbb{R}^n$. For simplicity, $\{1, 2, \cdots, N\}$ is denoted as $[[N]]$. The time-varying game is denoted as $\mathcal{G}_t := \{\mathcal{N}, \mathcal{Z}, f_t(z)\}$ where $\mathcal{N} := [[N]]$ denotes the set of players, $\mathcal{Z}$ is the set of actions and also a closed convex set, and $f_t(z)$ is the vector of cost functions of the players at time $t$.

Table 1: Comparison of convergence rates of several algorithms in time-invariant and two types of time-varying games. Alg stands for Algorithm.

| ALG | TIME-INVARIANT | FAST CONVERGENT PERTURBED | PERIODIC |
|---|---|---|---|
| PEG | $O(1/\sqrt{T})$ (CAI ET AL. (2022)) | $O(1/\sqrt{T})$ (CHEN & YU (2025)) | NONE (FENG ET AL. (2023)) |
| EG | $O(1/\sqrt{T})$ (CAI ET AL. (2022)) | $O(1/\sqrt{T})$ (CHEN & YU (2025)) | OPEN |
| RG | $O(1/\sqrt{T})$ (CAI & ZHENG (2023)) | $O(1/\sqrt{T})$ (THEOREM 2) | NONE (THEOREM 1) |
| ARG | $O(1/T)$ (CAI & ZHENG (2023)) | $O(1/T)$ (THEOREM 4) | NONE (THEOREM 3) |
| OG | $O(1/\sqrt{T})$ (CAI & ZHENG (2023)) | $O(1/\sqrt{T})$ (THEOREM 6) | $O(1/\sqrt{T})$ (THEOREM 6) |
| VOG | $O(1/\sqrt{T})$ (SAME AS OG) | OPEN | NONE (THEOREM 5) |

## 2.2 MODEL OF TIME-VARYING GAMES

In this part, we establish the time-varying game model, in which each player faces different cost functions at different times. With such a model, we can deal with the exogenous disturbance and the endogenous varying property. To begin with, we construct the framework of cost functions. For regularity, our paper only discusses continuously differentiable cost functions. For any time-varying game involved, the cost function of such a game at time $t$ is denoted as $f_t(z)$, and $F_t^{(i)}(z)$ denotes its gradient for Player $i$, i.e., $F_t^{(i)}(z) = \nabla f_t^{(i)}(z)$. By putting all players together, we define $F_t(z) = (F_t^{(1)}(z), \cdots, F_t^{(N)}(z))$. Specifically, we study the following two kinds of time-varying games: periodic games and convergent perturbed games.

Our time-varying games satisfy the following conditions:

1. At each time $t = 1, 2, \cdots$, each player selects their action $z^{(i)}$, $i \in \mathcal{N}$.
2. Each player pays a cost associated with $\mathcal{G}_t$ and selects their strategy based on the gradient of the set of actions of all the players, i.e., with the RG algorithm introduced in this paper.
3. All players implement their strategies and the sequence of games continues.

The essential elements of the machine learning model for the time-varying games include:

1. the game $\mathcal{G}_t$ played by all players at each time;
2. the gradients of time-varying cost functions of the players, $F_t^{(1)}(z), \cdots, F_t^{(N)}(z)$, at each time;
3. the algorithm for players to select their actions based on the gradients at each time.

The following parts present the properties of the elements above in detail.

**Definition 1** (Periodic games). *A periodic game with period $\mathcal{T}$ is an infinite sequence of games $\{f_t\}_{t=0}^{\infty} \subset \mathbb{R}^n$, and $f_{t+\mathcal{T}} = f_t$ for all $t \geq 0$.*

**Definition 2** (Convergent perturbed games). *A convergent perturbed game is an infinite sequence of games where $\lim_{t\to\infty} f_t$ exists. For simplicity, $\lim_{t\to\infty} f_t$ is denoted as $f_\infty$ and $g_t := f_t - f_\infty$. $\lim_{t\to\infty} g_t = 0$ by definition.*

Throughout the paper, sequences of time-varying smooth and monotone games with their limits being $C^2$ smooth cost functions are applied in our model. Besides, cost functions in time-invariant cases and their limits in time-varying cases are required to have monotone gradients. Following is the precise requirement of smoothness and monotonicity.

**Definition 3.** *((Rosen (1965); Cai et al. (2022); Chen & Yu (2025))) A game at a certain time is $C^2$ smooth and monotone if its cost function is $C^2$ smooth and monotone at the time, i.e., $F(z)$ is L-Lipschitz and monotone, which means that both $\|F(z) - F(z')\| \leq L\|z - z'\|$ and $\langle F(z) - F(z'), z - z' \rangle \geq 0$, where $L \geq 0$, holds.*

*A convergent perturbed game is $C^2$ smooth and monotone if the sequence of cost functions converges to a monotone function with Lipschitz gradient, i.e., $F_\infty(z) := \lim_{t\to\infty} F_t(z)$ is monotone and*

*L-Lipschitz, and a periodic game is $C^2$ smooth and monotone if the cost function $F_t(z)$ at each time $t$ is a monotone function with Lipschitz gradient.*

In the convergent perturbed games, another important assumption is that the difference between the time-varying cost function and its limit $f_t - f_\infty =: g_t$ satisfies the well-known assumption of bounded accumulated perturbations (BAP) (Benzaid & Lutz (1987); Saber Elaydi & Kamiyama (1999); Elaydi & Györi (1995)). Following are our settings for the BAP assumption.

**Assumption 1** (BAP assumption). *Denote $G_t := \nabla g_t$. $\mathcal{Z}$ is bounded and $\sum_{t=0}^{\infty} \max \|G_t\| < \infty$.*

### 2.3 NASH EQUILIBRIA IN TIME-VARYING GAMES

The most prevalent concept of solutions in game theory is the Nash equilibrium. The general definition of Nash equilibrium of a game $\mathcal{G}$ is an action $z^* \in \mathcal{Z}$ such that for each player $i$, it holds that $f^{(i)}(z^*) \leq f^{(i)}(z^{(i)}, z^{*(-i)})$ for any $z^{(i)} \in \mathcal{Z}^{(i)}$, where $\mathcal{Z}^{(i)}$ means the action set of Player $i$. By the general definition of Nash equilibrium, we obtain the following sufficient and necessary condition for an action to be a Nash equilibrium in our problems.

**Lemma 1.** *(Facchinei & Pang (2007)) For a time-invariant game $\mathcal{G}$ with monotone $F(z)$, an action $z^*$ is a Nash equilibrium if and only if $\langle F(z^*), z^* - z \rangle \leq 0, \forall z \in \mathcal{Z}$.*

Lemma 1 above means that solving the Nash equilibrium of a game in our paper requires solving the variational inequality in this lemma. Based on Lemma 1, for periodic games with common Nash equilibria and convergent perturbed games, we define the Nash equilibrium as follows.

**Definition 4.** *For a periodic game $\mathcal{G}$ with monotone $F_t(z)$ and an action $z^*$ satisfying $\langle F_t(z^*), z^* - z \rangle \leq 0, \forall z \in \mathcal{Z}$, its Nash equilibrium is defined as such $z^*$.*

**Definition 5.** *For a convergent perturbed game $\mathcal{G}$ with monotone $F_\infty(z)$, its Nash equilibrium is defined as an action $z^*$ satisfying $\langle F_\infty(z^*), z^* - z \rangle \leq 0, \forall z \in \mathcal{Z}$.*

### 2.4 TANGENT RESIDUAL

We apply the tangent residual proposed in (Cai et al. (2022)) as the measure of error in our paper. For time-varying games, we modify the definition of the tangent residual as follows.

**Definition 6.** *Denote the tangent residual of $f(z)$ involved in the game $\mathcal{G}$ as $r_{F,\mathcal{Z}}^{tan}$ and denote $N_{\mathcal{Z}}(z)$ as the normal cone of $z$ for the set $\mathcal{Z}$ (for simplicity $\mathcal{Z}$ may be omitted). The tangent residual of a periodic game with a common Nash equilibrium is defined as $r_{\mathcal{G}_i,\mathcal{Z}}^{tan} = \min_{c \in N(z)} \|F_i(z) + c\|$, where $i \in [[\mathcal{T}]]$ is the iteration number and $\mathcal{T}$ is the period in Definition 1, while the tangent residual of a convergent perturbed game is defined as $r_{\mathcal{G},\mathcal{Z}}^{tan} = \min_{c \in N(z)} \|F_\infty(z) + c\|$.*

### 2.5 MACHINE LEARNING ALGORITHMS IN GAMES

We introduce the following algorithms for our game problems: reflected gradient (RG) algorithm, accelerated reflected gradient (ARG) algorithm and optimistic gradient (OG) algorithm. Their pseudo-code implementations are shown in Algorithms 1, 2 and 3. All those three algorithms are variants of inexact proximal point algorithms (He & Yuan (2012); Zhang et al. (2025)), like EG and PEG discussed in existing results.

**Remark 2.** *In all three following learning algorithms, we do not limit the relationship between two initial points (Hsieh et al. (2019); Feng et al. (2023)) to avoid reducing generality of our results.*

---

**Algorithm 1** Reflected gradient algorithm (Malitsky (2015))

---

**Input:** Step size: $\eta > 0$; gradient of cost function: $F_t(x)$;
1: initialize $z_0$, $z_{-1}$
2: **for** $t = 0, 1, 2, 3, \cdots$ **do**
3:     play $z_t \in \mathcal{Z}$
4:     set $z_{t+1/2} = 2z_t - z_{t-1}$
5:     set $z_{t+1} = \Pi_{\mathcal{Z}}(z_t - \eta F_t(z_{t+1/2}))$
6: **end for**

---

---

**Algorithm 2** Accelerated reflected gradient algorithm (Cai & Zheng (2023))

---

**Input:** Step size: $\eta > 0$; gradient of cost function: $F_t(x)$;
1: initialize $z_0$, $z_{1/2}$
2: **for** $t = 0, 1, 2, 3, \cdots$ **do**
3:    play $z_t \in \mathcal{Z}$
4:    set $z_{t+1/2} = 2z_t - z_{t-1} + \frac{1}{t+1}(z_0 - z_t) - \frac{1}{t}(z_0 - z_{t-1})$ if $t \neq 0$
5:    set $z_{t+1} = \Pi_{\mathcal{Z}}(z_t - \eta F_t(z_{t+1/2}) + \frac{1}{t+1}(z_0 - z_t))$
6: **end for**

---

**Algorithm 3** Optimistic gradient algorithm (Hsieh et al. (2019); Mokhtari et al. (2020a;b); Daskalakis et al. (2018))

---

**Input:** Step size: $\eta > 0$; gradient of cost function: $F_t(x)$;
1: initialize $z_0$, $z_{-1/2}$
2: **for** $t = 0, 1, 2, 3, \cdots$ **do**
3:    play $z \in \mathcal{Z}$
4:    set $z_{t+1/2} = \Pi_{\mathcal{Z}}(z_t - \eta F_t(z_{t-1/2}))$
5:    set $z_{t+1} = z_{t+1/2} + \eta F_t(z_{t-1/2}) - \eta F_t(z_{t+1/2})$
6: **end for**

---

## 3    CONVERGENCE RESULTS OF THE RG ALGORITHM

In this section, we prove that the RG algorithm does not necessarily converge for a periodic game even if that game is a two-player zero-sum game, though it reaches a convergence result of $O(1/\sqrt{T})$ with a fast converging perturbation. To illustrate Theorem 1 and Theorem 2 in this section, numerical examples based on examples of bilinear games in (Feng et al. (2023)) are shown in Section I.

### 3.1   PERIODIC CASE

In this section, we show a negative result: the RG algorithm does not necessarily converge in monotone periodic games, even if the step size is small. To illustrate this negative result, we provide a simple counterexample with a bilinear game inspired from (Feng et al. (2023)).

**Theorem 1.** *In the following two-player game $\mathcal{G}_t$, no $\eta > 0$ guarantees that the RG algorithm converges, since the RG algorithm diverges at an exponential rate in this example.*

$$z^{(1)} = x \in \mathbb{R}, z^{(2)} = y \in \mathbb{R}^2, f^{(1)} = x^T A_t y, f^{(2)} = -x^T A_t y, \mathcal{Z} = \mathbb{R} \times \mathbb{R}^2$$

$$A_t = \begin{cases} \begin{bmatrix} 1 & -1 \end{bmatrix}, t \text{ is odd} \\ \begin{bmatrix} -1 & 1 \end{bmatrix}, t \text{ is even} \end{cases} \tag{1}$$

*Proof sketch.* For the two-player bilinear game with the period of 2 above, we prove that $\mathcal{A}_t$ satisfying $\begin{bmatrix} x_t & y_t & x_{t+1} & y_{t+1} \end{bmatrix}^T = \mathcal{A}_t \begin{bmatrix} x_{t-1} & y_{t-1} & x_t & y_t \end{bmatrix}^T$ has eigenvalues greater than 1 so that $\|z_t\|$ increases exponentially with $\eta > 0$ in certain cases. Then, with $r^{tan}(z_t) = \|z_t\|$ in this game, we conclude that $r^{tan}(z_t)$ diverges with the RG algorithm in such cases, which results in the impossibility of the RG algorithm converging to the Nash equilibrium in such cases.

The full proof of Theorem 1 is deferred to Section C.

### 3.2   BAP CASE

In this part, we discuss the case of time-varying games for the RG algorithm under BAP assumption, i.e., Assumption 1. We show that $L_{G_t}$-smooth perturbation makes convergent perturbed games with a common Nash equilibrium satisfy Assumption 1 naturally.

**Lemma 2.** *If there exists a common Nash equilibrium $z^*$ for each time of a sequence of convergent perturbed monotone games and $G_t(z)$ is $L_{G_t}$-Lipschitz with $\sum_{t=0}^{\infty} L_{G_t} < \infty$ and $G_t(z^*) = 0$, then there exists a bounded set $\mathcal{Z}_L$ satisfying $z_t \in \mathcal{Z}_L$. Consequently, Assumption 1 is satisfied.*

In the following theorem, we show that the RG algorithm is robust to perturbation satisfying Assumption 1.

**Theorem 2.** *For the convergent perturbed game with a bounded action set and its limit having a L-Lipschitz gradient, if $\eta \in (0, \frac{1}{(\sqrt{2}+1)L})$, then the RG algorithm converges and the convergence rate of the tangent residual is* $\max \left\{ O(1/\sqrt{T}), O\left(\sqrt{\sum_{t=T/2}^{\infty} \max \|G_t\|}\right) \right\}$ *under Assumption 1.*

*Proof sketch.* We construct a potential function and show that it increases slowly or does not increase between two consecutive iterates. We prove that the best-iterate convergence rate of the RG algorithm is small, i.e., $\forall T \geq 1$, there exists one iterate $t^* \in [[T]]$ such that our potential function at $t^*$ is small. We combine the above steps to show that the last iterate has the same convergence guarantee as the best iterate and show that the last-iterate convergence rate is $O(1/\sqrt{T})$ under Assumption 1.

The full proof of Theorem 2 is deferred to Section D and the proof of Lemma 2 is included there in Section D.3.

# 4 CONVERGENCE RESULTS OF THE ARG ALGORITHM

In this section, we prove the following results: the ARG algorithm does not necessarily converge for a periodic game even if that game is a two-player zero-sum game. For completeness, we also show the ARG algorithm reaches a convergence result of $O(1/T)$ with a fast converging perturbation and still converges with a perturbation converging slower. To illustrate Theorem 3 and Theorem 4 in this section, numerical examples based on bilinear games in (Feng et al. (2023)) are shown in Section I.

## 4.1 PERIODIC CASE

In this section, we show a negative result: the ARG algorithm does not necessarily converge in monotone periodic games, even if the step size is small. To illustrate this negative result, we provide a simple counterexample with a bilinear game inspired from (Feng et al. (2023)).

**Theorem 3.** *In the two-player game (1) shown in Theorem 1, no $\eta > 0$ guarantees the ARG algorithm converges, since the ARG algorithm diverges at an exponential rate in the example provided.*

*Proof sketch.* For the two-player bilinear game with the period of 2 above, we prove that for a $t$ large enough, $\mathcal{A}_t$ satisfying $[x_0 \quad y_0 \quad x_t \quad y_t \quad x_{t+1} \quad y_{t+1}]^T = \mathcal{A}_t [x_0 \quad y_0 \quad x_{t-1} \quad y_{t-1} \quad x_t \quad y_t]^T$ converges to a constant matrix with eigenvalues greater than 1, making $\|z_{t+T}\|$ is approximately equal to $C^T \|z_t\|$ with $\eta > 0$ and $C > 1$. Then, with $r^{tan}(z_t) = \|z_t\|$ in this game, we show that $r^{tan}(z_t)$ diverges with the ARG algorithm.

The full proof of Theorem 3 is deferred to Section E.

## 4.2 BAP CASE

In this part, we show the ARG algorithm is robust to perturbation with a fast convergence rate.

**Theorem 4.** *For the convergent perturbed game with a bounded action set and its limit having a L-Lipschitz gradient, if $\eta \in (0, \frac{1}{\sqrt{24}L})$ and $\sum_{t=0}^{\infty} t^2 \|G_t\| < \infty$, the ARG algorithm converges at the rate $O(1/T)$. Under Assumption 1, the tangent residual with the ARG algorithm is* $\max \left\{ O(1/\sqrt{T}), O\left(\sqrt{\sum_{t=1}^{T} \max \|G_t\|}\right) \right\}$. *This result can be extended to $\rho \in [-\frac{1}{60}, 0]$ and $\langle F_t(z) + N(z) - F_t(z') - N(z'), z - z' \rangle \geq \rho \|F_t(z) + N(z) - F_t(z') - N(z')\|^2, \forall z, z' \in \mathcal{Z}$ if $\frac{1}{2} - (12 - \frac{4\rho}{\eta})\eta^2 L^2 + \frac{2\rho}{\eta} \geq 0$.*

*Proof sketch.* We only need to prove the extended result. We first construct a potential function and show that it is approximately non-increasing. Then, we prove that it is upper bounded by a term independent of time and prove that the such function at step $t$ is $O(t^2 r^{tan}(z_t)^2)$ if $\sum_{t=0}^{\infty} t(t+1)\|G_t\| < \infty$ and is $O(tr^{tan}(z_t)^2)$ under Assumption 1. Then, we conclude that the convergence rate of the ARG algorithm is $O(1/T)$ if $\sum_{t=0}^{\infty} t(t+1)\|G_t\| < \infty$ and is $O(1/\sqrt{T})$ under Assumption 1.

The full proof of Theorem 4 is deferred to Section F.

## 5 CONVERGENCE RESULTS OF THE OG ALGORITHM

In this section, we show that the OG algorithm converges for any $L$-smooth game, including any periodic game and convergent perturbed game, with a common Nash equilibrium, while surprisingly, a variant of the OG algorithm may diverge exponentially even if that game is a periodic two-player zero-sum game with a common Nash equilibrium. To illustrate Theorem 5 and Theorem 6 in this section, numerical examples based on bilinear games in (Feng et al. (2023)) are shown in Section I.

### 5.1 PERIODIC CASE OF A VARIANT OF THE OG ALGORITHM

In this part, we show a surprising fact that it is not necessarily possible for a natural variant of the OG algorithm with still only one calculation on the gradient and the projection to converge with periodic monotone games, since it is almost equivalent to the RG algorithm in two-player bilinear games.

**Theorem 5.** *If $F_t$ in Line 5 of the OG algorithm (Algorithm 3) is modified to $F_{t+1}$, then the modified OG algorithm for $z_{t+1/2}$ is equivalent to the RG algorithm for $z_t$ in two-player bilinear games, including (1) shown in Theorem 1. Alternatively, in periodic monotone games, no $\eta > 0$ guarantees that the tangent residual converges with the modified OG algorithm, since the modified OG algorithm diverges at an exponential rate in the example provided.*

*Proof sketch.* We construct a two-player bilinear game with the period of 2 and prove that for $t = 2k - 1$ there exists a initial point $z_0$ and $z_{-1/2}$ making $\|z_t\| \geq C\lambda^k$ with $\eta, C > 0$ and $\lambda > 1$. Then, with $r^{tan}(z_t) = \|z_t\|$ in this game, we show that $r^{tan}(z_t)$ diverges with the variant of the OG algorithm.

The full proof of Theorem 5 is deferred to Section G.

### 5.2 TIME-VARYING CASE OF THE STANDARD OG ALGORITHM APPLICABLE TO PERIODIC CASE

In this part, we show that the weak convergence of the standard OG algorithm is highly robust to the changing cost function as long as $L_t$ is bounded.

**Theorem 6.** *If there exists a common Nash equilibrium $z^*$ and the game at each time $t$ is monotone and $L_t$-Lipschitz with the step size $\eta$ satisfying $\eta \in (0, \frac{1}{2L_{\max}})$ for the OG algorithm, then $\forall T \geq 1$, under the Assumption*

$$\min_{t \in [[T]]} r^{tan}_{F, \mathcal{Z}}(z_{t+\frac{1}{2}}) \leq \min_{t \in [[T]]} \frac{\|z_{t+1} - z_t\|}{\eta} = O\left(\frac{1}{\sqrt{T}}\right) \tag{2}$$

*where $L_{\max} = \max_{t \in [[T]]} L_t$ and $\langle N(z), z - z^* \rangle \geq \rho \|N(z)\|^2$. This result can be extended to $\langle F_t(z) + N(z) - F_t(z') - N(z'), z - z' \rangle \geq \rho \|F_t(z) + N(z) - F_t(z') - N(z')\|^2, \forall z, z' \in \mathcal{Z}$ with $\rho \in (-\frac{1}{12\sqrt{3}L_{\max}}, 0]$ if $\frac{1}{2} + \frac{2\rho}{\eta} - 2\eta^2 L_{\max}^2 > 0$. Note that $L_{\max}$ exists for periodic games since there are at most $\mathcal{T}$ different games all the time so that $L_t$ is in a limited set and there exists a greatest $L_t$.*

The proof of Theorem 6 is deferred to Section H.

## 6 CONCLUSION

In this paper, we provide a surprising result that the RG, ARG and even modified OG algorithms may diverge at an exponential rate for periodic games even if only two players are involved where the standard OG algorithm converges. We also prove that the RG and ARG algorithms with constant step sizes converge to the Nash equilibrium in time-varying multi-player convergent perturbed monotone games with bounded action sets at satisfactory rates. This is comparable to those algorithms and related algorithms for both time-invariant and time-varying games.

There remain some interesting future research topics. One is whether a last-iterate convergence rate can be established for the OG algorithm. Another is whether and when the RG, OG, and ARG algorithms behave similarly for stochastic games no less general than those in this paper.

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

## A  USAGE OF LARGE LANGUAGE MODELS (LLMs) IN THE PAPER

LLMs are applied to revise the potential grammatical mistakes and improve wording of sentences. No significant contents are generated by LLMs.

## B  ADDITIONAL LEMMAS

In our proof, the following lemmas are necessary.

**Lemma 3.** *(Cai & Zheng (2023))* $\forall x_i, y_j, u_2, u_4 \in \mathbb{R}^n$ *with* $i = 0, 1, 2, 3, 4, j = 1, 2, 3, 4$ *and* $k \geq 1, q \in (0, 1)$, *if* $x_3 = x_2 - y_1 - u_2$ *and* $x_4 = x_2 - y_3 - u_4$, *then there exists*

$$\|y_2 + u_2\|^2 + \|y_2 - y_1\|^2 - \|y_4 + u_4\|^2 - \|y_4 - y_3\|^2 - 2\langle y_4 - y_2, x_4 - x_2 \rangle$$

$$- 2\left(\frac{1}{4}\|x_4 - x_3\|^2 - \|y_4 - y_3\|^2\right) - 2\langle u_4 - u_2, x_4 - x_2 \rangle \tag{3}$$

$$= \left\|\frac{x_3 - x_4}{2} + y_1 - y_2\right\|^2 + \left\|\frac{x_3 + x_4}{2} - x_2 + y_2 + u_2\right\|^2$$

**Lemma 4.** *(Cai & Zheng (2023))* $\forall x_i, y_j, u_2, u_4 \in \mathbb{R}^n$ *with* $i = 0, 1, 2, 3, 4, j = 1, 2, 3, 4$ *and* $k \geq 1, q \in (0, 1)$, *if* $x_3 = x_2 - y_1 - u_2 + \frac{1}{k+1}(x_0 - x_2)$ *and* $x_4 = x_2 - y_3 - u_4 + \frac{1}{k+1}(x_0 - x_2)$ *then there exists*

$$\frac{k(k+1)}{2}\left(\|y_2 + u_2\|^2 + \|y_2 - y_1\|^2\right) + k\langle y_2 + u_2, x_2 - x_0 \rangle - \frac{(k+1)(k+2)}{2}\left(\|y_4 + u_4\|^2 + \|y_4 - y_3\|^2\right)$$

$$- (k+1)\langle y_4 + u_4, x_4 - x_0 \rangle - k(k+1)\langle y_4 + u_4 - y_2 - u_2, x_4 - x_2 \rangle - \frac{k(k+1)}{4q}\left\langle q\|x_4 - x_3\|^2 - \|y_4 - y_3\|^2 \right\rangle$$

$$= \frac{k(k+1)}{4}\|u_4 - u_2 + y_1 - 2y_2 + y_3\|^2 + \left(\frac{(1-4q)k - 4q}{4q}(k+1)\right)\|y_3 - y_4\|^2 + (k+1)\langle y_3 - y_4, y_4 + u_4 \rangle$$

$$\tag{4}$$

**Lemma 5.** *(Little et al. (2022)) If $\{a_n\}$ is a sequence of non-negative numbers, then the series $\sum_{j=1}^{\infty} a_j$ and the product $\prod_{j=1}^{\infty}(1+a_j)$ either both converge or both diverge.*

## C  MISSING PROOFS OF THEOREM 1

In periodic games, each period as a whole is time-invariant (Franke & Selgrade (2003), Section 3). We take advantage of this property in our analysis.

$$z_{t+\frac{1}{2}} = 2z_t - z_{t-1} = \begin{bmatrix} 2x_t - x_{t-1} \\ 2y_t - y_{t-1} \end{bmatrix} \tag{5}$$

Hence, we have

$$z_{t+1} = z_t - \eta F_t(z_{t+\frac{1}{2}}) = z_t - \eta \begin{bmatrix} A_t(2y_t - y_{t-1}) \\ -A_t^T(2x_t - x_{t-1}) \end{bmatrix}$$

$$= \begin{bmatrix} x_t - 2\eta A_t y_t + \eta A_t y_{t-1} \\ y_t + 2\eta A_t^T x_t - \eta A_t^T x_{t-1} \end{bmatrix} \tag{6}$$

Then, there exists

$$\begin{bmatrix} x_t \\ y_t \\ x_{t+1} \\ y_{t+1} \end{bmatrix} = \begin{bmatrix} 0 & 0 & I & 0 \\ 0 & 0 & 0 & I \\ 0 & \eta A_t & I & -2\eta A_t \\ -\eta A_t^T & 0 & 2\eta A_t^T & I \end{bmatrix} \begin{bmatrix} x_{t-1} \\ y_{t-1} \\ x_t \\ y_t \end{bmatrix} \tag{7}$$

Denote $\begin{bmatrix} 0 & 0 & I & 0 \\ 0 & 0 & 0 & I \\ 0 & \eta A_t & I & -2\eta A_t \\ -\eta A_t^T & 0 & 2\eta A_t^T & I \end{bmatrix}$ as $P_t$. We have

$$P_t = \begin{cases} \begin{bmatrix} 0 & 0 & 0 & 1 & 0 & 0 \\ 0 & 0 & 0 & 0 & 1 & 0 \\ 0 & 0 & 0 & 0 & 0 & 1 \\ 0 & \eta & -\eta & 1 & -2\eta & 2\eta \\ -\eta & 0 & 0 & 2\eta & 1 & 0 \\ \eta & 0 & 0 & -2\eta & 0 & 1 \end{bmatrix}, & t \text{ is odd} \\[4em] \begin{bmatrix} 0 & 0 & 0 & 1 & 0 & 0 \\ 0 & 0 & 0 & 0 & 1 & 0 \\ 0 & 0 & 0 & 0 & 0 & 1 \\ 0 & -\eta & \eta & 1 & 2\eta & -2\eta \\ \eta & 0 & 0 & -2\eta & 1 & 0 \\ -\eta & 0 & 0 & 2\eta & 0 & 1 \end{bmatrix}, & t \text{ is even} \end{cases} \tag{8}$$

Without loss of generality, assume $t$ is odd. Let $\mathcal{A}_{t,t+1} = P_{t+1} P_t$. Then,

$$\begin{aligned} &\det(\lambda I - \mathcal{A}_{t,t+1}) \\ =& 64\eta^4\lambda^4 - 96\eta^4\lambda^3 + 36\eta^4\lambda^2 - 4\eta^4\lambda - 16\eta^2\lambda^5 \\ &+ 36\eta^2\lambda^4 - 24\eta^2\lambda^3 + 4\eta^2\lambda^2 + \lambda^6 - 3\lambda^5 + 3\lambda^4 - \lambda^3 \\ =& \lambda(\lambda - 1)(\lambda^4 - (16\eta^2 + 2)\lambda^3 + (64\eta^4 + 20\eta^2 + 1)\lambda^2 \\ &- (32\eta^4 + 4\eta^2)\lambda + 4\eta^4) \\ =& \lambda(\lambda - 1)(\lambda^2 - (8\eta^2 + 1)\lambda + 2\eta^2)^2 \\ =& \left(\lambda - \left(\frac{8\eta^2 + 1 + \sqrt{64\eta^4 + 8\eta^2 + 1}}{2}\right)\right)^2 \\ &\left(\lambda - \left(\frac{8\eta^2 + 1 - \sqrt{64\eta^4 + 8\eta^2 + 1}}{2}\right)\right)^2 \lambda(\lambda - 1) \end{aligned} \tag{9}$$

Then, $\mathcal{A}_{t,t+1}$ has an eigenvalue $\lambda_{1,2} = \frac{8\eta^2 + 1 + \sqrt{64\eta^4 + 8\eta^2 + 1}}{2} > 1$. Since step size $\eta > 0$, the spectral radius of matrix $\mathcal{A}_{t,t+1}$ is no less than $\lambda_{1,2}$, and then greater than 1. Denote the corresponding eigenvector of $\lambda_{1,2}$ for $\mathcal{A}_{t,t+1}$ as $v$. Then, we obtain

$$\left\| \left( \prod_{i=0}^{k} \mathcal{A}_{2i,2i+1} \right) v \right\| = \left\| \lambda_{1,2}^{k+1} v \right\| \tag{10}$$

Since the action set is $\mathbb{R}^3$, we have

$$r_t^{tan}(z) = \|F_t(z)\| = \left\| \begin{bmatrix} A_t y \\ -A_t^T x \end{bmatrix} \right\| \tag{11}$$

Hence, $\forall t \geq 1$,

$$\left\| r_{\mathcal{G}_t, \mathcal{Z}}^{tan}(z) \right\| = \left\| \begin{bmatrix} A_t y \\ -A_t^T x \end{bmatrix} \right\| = \|z\| \tag{12}$$

Since $\left\| \lambda_{1,2}^k v \right\| = \|\lambda_{1,2}\|^k \|v\|$ grows exponentially, $\|z_{2k}\|$ does not converge. This means that the tangent residuals of $\mathcal{G}$ with the RG algorithm diverge exponentially. This completes the proof.

## D   Missing proofs of Theorem 2

To prove Theorem 2, we apply a potential function argument. We first show that the potential function is approximately non-increasing and then prove that it is upper bounded by a term independent of $T$. As the potential function at step $t$ is also $O(t) r^{tan}(z_t)^2$, we conclude that the RG algorithm converges at the rate of $\max \left\{ O\left(\frac{1}{\sqrt{T}}\right), O\left(\sqrt{\sum_{t=T/2}^{\infty} \max \|G_t\|}\right) \right\}$.

### D.1   Approximate monotonicity of the potential function

To construct a potential function, we denote

$$c_{t+1} = \frac{z_t - \eta F_t(z_{t+\frac{1}{2}}) - z_{t+1}}{\eta}, \forall t \geq 0 \tag{13}$$

Note that according to the update rules of RG, $z_{t+1} = \Pi_{\mathcal{Z}}[z_t - \eta F_t(z_{t+1/2})]$. This means $c_{t+1} \in N_{\mathcal{Z}}(z_{t+1})$. To our analysis on the RG algorithm, the potential function $P_t$ is applied, which is defined as

$$P_t = \|F_{t-1}(z_t) + c_t\|^2 + \left\| F_{t-1}(z_t) - F_{t-1}(z_{t-\frac{1}{2}}) \right\|^2, \forall t \geq 1 \tag{14}$$

**Lemma 6.** *With the same settings as Theorem 2, $\forall t \geq 1$, $P_t - P_{t+1} \geq \left(-\frac{2}{\eta} - 8L\right) D_{\mathcal{Z}} \max \|G_t\| - \frac{2}{\eta} D_{\mathcal{Z}} \max \|G_{t-1}\| - 8 \max \|G_t\|^2$.*

*Proof.* We only need to prove that the sum of $P_t - P_{t+1} + \left(\frac{2}{\eta} + 8L\right) D_{\mathcal{Z}} \max \|G_t\| + \frac{2}{\eta} D_{\mathcal{Z}} \max \|G_{t-1}\| + 8 \max \|G_t\|^2$ and some non-positive terms is approximately non-negative. Since $F_\infty$ is monotone, we have

$$-2\langle \eta F_\infty(z_{t+1}) - \eta F_\infty(z_t), z_{t+1} - z_t \rangle \leq 0 \tag{15}$$

which means

$$-2\langle \eta F_t(z_{t+1}) - \eta F_{t-1}(z_t), z_{t+1} - z_t \rangle + 2\langle \eta G_t(z_{t+1}) - \eta G_{t-1}(z_t), z_{t+1} - z_t \rangle \leq 0 \tag{16}$$

Since $F_\infty$ is $L$-Lipschitz and $0 < \eta < \frac{1}{(1+\sqrt{2})L} < \frac{1}{2L}$, we have

$$\frac{1}{4} \left\| z_{t+1} - z_{t+\frac{1}{2}} \right\|^2 - \left\| \eta F_\infty(z_{t+1}) - \eta F_\infty(z_{t+\frac{1}{2}}) \right\|^2 \geq 0 \tag{17}$$

Since

$$\left\| \eta F_t(z_{t+1}) - \eta F_t(z_{t+\frac{1}{2}}) \right\|^2 - \left\| \eta G_t(z_{t+1}) - \eta G_t(z_{t+\frac{1}{2}}) \right\|^2$$

$$- 2\left\langle \eta F_\infty(z_{t+1}) - \eta F_\infty(z_{t+\frac{1}{2}}), \eta G_t(z_{t+1}) - \eta G_t(z_{t+\frac{1}{2}}) \right\rangle - \left\| \eta F_\infty(z_{t+1}) - \eta F_\infty(z_{t+\frac{1}{2}}) \right\|^2$$

$$= \left\| \eta F_t(z_{t+1}) - \eta F_t(z_{t+\frac{1}{2}}) \right\|^2 - \left( \left\| \eta G_t(z_{t+1}) - \eta G_t(z_{t+\frac{1}{2}}) \right\| + \left\| \eta F_\infty(z_{t+1}) - \eta F_t(z_{t+\frac{1}{2}}) \right\| \right)^2$$

$$\leq 0 \tag{18}$$

we have

$$- 2\left( \frac{1}{4} \left\| z_{t+1} - z_{t+\frac{1}{2}} \right\|^2 - \left\| \eta F_t(z_{t+1}) - \eta F_t(z_{t+\frac{1}{2}}) \right\|^2 \right)$$

$$\leq - 2\left( - \left\| \eta G_t(z_{t+1}) - \eta G_t(z_{t+\frac{1}{2}}) \right\|^2 - 2\left\langle \eta F_\infty(z_{t+1}) - \eta F_\infty(z_{t+\frac{1}{2}}), \eta G_t(z_{t+1}) - \eta G_t(z_{t+\frac{1}{2}}) \right\rangle \right) \tag{19}$$

so that

$$- 2\left( \frac{1}{4} \left\| z_{t+1} - z_{t+\frac{1}{2}} \right\|^2 - \left\| \eta F_t(z_{t+1}) - \eta F_t(z_{t+\frac{1}{2}}) \right\|^2 \right) - 2\left\| \eta G_t(z_{t+1}) - \eta G_t(z_{t+\frac{1}{2}}) \right\|^2$$

$$- 4\left\langle \eta F_\infty(z_{t+1}) - \eta F_\infty(z_{t+\frac{1}{2}}), \eta G_t(z_{t+1}) - \eta G_t(z_{t+\frac{1}{2}}) \right\rangle \tag{20}$$

$$\leq 0$$

By the definition of $c_t$ and $c_{t+1}$, we have $c_{t+1} \in N_{\mathcal{Z}}(z_{t+1})$ and $c_t \in N_{\mathcal{Z}}(z_t)$. Since the normal cone operator $N_{\mathcal{Z}}$ is maximally monotone, we have

$$-2\langle \eta c_{t+1} - \eta c_t, z_{t+1} - z_t \rangle \leq 0 \tag{21}$$

$\square$

We use the following equivalent formations of $z_{t+1/2}$ and $z_{t+1}$.

$$z_{t+\frac{1}{2}} = 2z_t - z_{t-1} = z_t - (z_{t-1} - z_t) = z_t - \eta F_{t-1}(z_{t-\frac{1}{2}}) - \eta c_t$$

$$z_{t+1} = \Pi_{\mathcal{Z}}\left[ z_t - \eta F_t(z_{t+\frac{1}{2}}) \right] = z_t - \eta F_t(z_{t+\frac{1}{2}}) - \eta c_{t+1} \tag{22}$$

According to Lemma 3, by replacing $x_k$ with $z_{t-1+\frac{k}{2}}$, $y_k$ with $\eta F_{t-1}(z_{t-1+\frac{k}{2}})$, $u_2$ with $\eta c_t$, and $u_4$ with $\eta c_{t+1}$, since both $x_3 = x_2 - y_1 - u_2$ and $x_4 = x_2 - y_3 - u_4$ hold true, we have

$$\eta^2(P_t - P_{t+1}) + \text{LHS of Inequality (16)} + \text{LHS of Inequality (20)} + \text{LHS of Inequality (21)}$$

$$= \left\| \frac{z_{t+\frac{1}{2}} - z_{t+1}}{2} + \eta F_{t-1}(z_{t-\frac{1}{2}}) - \eta F_{t-1}(z_t) \right\|^2 + \left\| \frac{z_{t+\frac{1}{2}} + z_{t+1}}{2} - z_t + \eta F_{t-1}(z_t) + \eta c_t \right\|^2$$

$$+ 2\langle \eta G_t(z_{t+1}) - \eta G_{t-1}(z_t), z_{t+1} - z_t \rangle - 2\left\| \eta G_t(z_{t+1}) - \eta G_t(z_{t+\frac{1}{2}}) \right\|^2$$

$$- 4\left\langle \eta F_\infty(z_{t+1}) - \eta F_\infty(z_{t+\frac{1}{2}}), \eta G_t(z_{t+1}) - \eta G_t(z_{t+\frac{1}{2}}) \right\rangle \tag{23}$$

Since $\mathcal{Z}$ is bounded and $z_t, z_{t+1} \in \mathcal{Z}$ there exists $D_{\mathcal{Z}} > 0$ so that $\| z_t - z_{t+1} \| \leq D_{\mathcal{Z}}$, we have

$$2\langle \eta G_t(z_{t+1}) - \eta G_{t-1}(z_t), z_{t+1} - z_t \rangle - 2\left\| \eta G_t(z_{t+1}) - \eta G_t(z_{t+\frac{1}{2}}) \right\|^2$$

$$- 4\left\langle \eta F_\infty(z_{t+1}) - \eta F_\infty(z_{t+\frac{1}{2}}), \eta G_t(z_{t+1}) - \eta G_t(z_{t+\frac{1}{2}}) \right\rangle$$

$$\geq - 2\eta D_{\mathcal{Z}}(\max \|G_t\| + \max \|G_{t-1}\|) - 8\eta^2 \max \|G_t\|^2$$

$$- 4\eta^2 \left\| F_\infty(z_{t+1}) - F_\infty(z_{t+\frac{1}{2}}) \right\| \left\| G_t(z_{t+1}) - G_t(z_{t+\frac{1}{2}}) \right\|$$

$$\geq - 2\eta D_{\mathcal{Z}}(\max \|G_t\| + \max \|G_{t-1}\|) - 8\eta^2 \max \|G_t\|^2 - 8L\eta^2 \left\| z_{t+1} - z_{t+\frac{1}{2}} \right\| \max \|G_t\|$$

$$\geq - 2\eta D_{\mathcal{Z}}(\max \|G_t\| + \max \|G_{t-1}\|) - 8\eta^2 \max \|G_t\|^2 - 8L\eta^2 D_{\mathcal{Z}} \max \|G_t\|$$

$$= (-2\eta - 8L\eta^2)D_{\mathcal{Z}} \max \|G_t\| - 2\eta D_{\mathcal{Z}} \max \|G_{t-1}\| - 8\eta^2 \max \|G_t\|^2 \tag{24}$$

Thus we conclude $P_t - P_{t+1} \geq (-\frac{2}{\eta} - 8L)D_{\mathcal{Z}} \max \|G_t\| - \frac{2}{\eta}D_{\mathcal{Z}} \max \|G_{t-1}\| - 8 \max \|G_t\|^2$.

## D.2  BEST-ITERATE CONVERGENCE OF THE ERROR FROM THE OPTIMAL SOLUTION

In this section, we show that for any $T \geq 1$, there exists $t^*$ satisfying $P_{t^*} = O(\frac{1}{T})$, which is implied by $\sum_{t=1}^{T} P_t = O(1)$. To prove this, we first prove $\sum_{t=1}^{T} \|z_{t+1/2} - z_t\|^2 = \sum_{t=1}^{T} \|z_t - z_{t-1}\|^2 = O(1)$ and then express $\sum_{t=1}^{T} P_t$ with these two quantities.

**Lemma 7.** *In the settings of Theorem 2, for any $T \geq 1$, we have*

$$\sum_{t=1}^{T} \left\| z_{t+\frac{1}{2}} - z_t \right\|^2 = \sum_{t=1}^{T} \|z_t - z_{t-1}\|^2 \leq \frac{H^2}{1 - (1 + \sqrt{2})\eta L} \tag{25}$$

*Proof.* By the update rules of the RG algorithm, we have $z_{t+1/2} = 2z_t - z_{t-1}$ so that $z_{t+1/2} - z_t = z_t - z_{t-1}$. Therefore, it is only necessary to prove the inequality for $\sum_{t=1}^{T} \|z_{t+1/2} - z_t\|^2$. With the assistance of the proof of Lemma 2 in Hsieh et al. (2019), $\forall t \geq 1$ and $p \in \mathcal{Z}$, we have

$$\|z_{t+1} - p\|^2$$
$$\leq \|z_t - p\|^2 + 2\langle \eta F_t(z_{t+\frac{1}{2}}) - (z_{t-1} - z_t), z_{t+\frac{1}{2}} - z_{t+1} \rangle - 2\eta\langle F_t(z_{t+\frac{1}{2}}), z_{t+\frac{1}{2}} - p \rangle \tag{26}$$
$$- \left\| z_{t+1} - z_{t+\frac{1}{2}} \right\|^2 - \left\| z_{t+\frac{1}{2}} - z_t \right\|^2$$

According to the update rule of the RG algorithm, we have

$$\langle z_t - (z_{t-1} - \eta F_{t-1}(z_{t-\frac{1}{2}})), z_t - z_{t-1} \rangle \leq 0 \tag{27}$$

$$\langle z_t - (z_{t-1} - \eta F_{t-1}(z_{t-\frac{1}{2}})), z_t - z_{t+1} \rangle \leq 0 \tag{28}$$

The sum of (27) and (28) is

$$\langle z_t - z_{t-1}, z_{t+\frac{1}{2}} - z_{t+1} \rangle \leq -\langle \eta F_{t-1}(z_{t-\frac{1}{2}}), z_{t+\frac{1}{2}} - z_{t+1} \rangle \tag{29}$$

Hence, we have

$$2\langle \eta F_t(z_{t+\frac{1}{2}}) - (z_{t-1} - z_t), z_{t+\frac{1}{2}} - z_{t+1} \rangle$$
$$\leq 2\langle \eta F_t(z_{t+\frac{1}{2}}) - \eta F_t(z_{t-\frac{1}{2}}), z_{t+\frac{1}{2}} - z_{t+1} \rangle$$
$$= 2\eta \left( \left\langle F_\infty(z_{t+\frac{1}{2}}) - F_\infty(z_{t-\frac{1}{2}}), z_{t+\frac{1}{2}} - z_{t+1} \right\rangle + \left\langle G_t(z_{t+\frac{1}{2}}) - G_{t-1}(z_{t-\frac{1}{2}}), z_{t+\frac{1}{2}} - z_{t+1} \right\rangle \right)$$
$$\leq 2\eta \left( L \left\| z_{t+\frac{1}{2}} - z_{t-\frac{1}{2}} \right\| \left\| z_{t+\frac{1}{2}} - z_{t+1} \right\| + \left\langle G_t(z_{t+\frac{1}{2}}) - G_{t-1}(z_{t-\frac{1}{2}}), z_{t+\frac{1}{2}} - z_{t+1} \right\rangle \right)$$
$$\leq 2\eta \left( L \left\| z_{t+\frac{1}{2}} - z_{t-\frac{1}{2}} \right\| \left\| z_{t+\frac{1}{2}} - z_{t+1} \right\| + \left\| G_t(z_{t+\frac{1}{2}}) - G_{t-1}(z_{t-\frac{1}{2}}) \right\| \left\| z_{t+\frac{1}{2}} - z_{t+1} \right\| \right) \tag{30}$$

By combining (26) and (30), we have

$$\|z_{t+1} - p\|^2$$
$$\leq \|z_t - p\|^2 + 2\eta L \left\| z_{t+\frac{1}{2}} - z_{t-\frac{1}{2}} \right\| \left\| z_{t+\frac{1}{2}} - z_{t+1} \right\| + 2\eta \left\langle G_t(z_{t+\frac{1}{2}}) - G_{t-1}(z_{t-\frac{1}{2}}), z_{t+\frac{1}{2}} - z_{t+1} \right\rangle$$
$$- 2\eta\langle F_t(z_{t+\frac{1}{2}}), z_{t+\frac{1}{2}} - p \rangle - \left\| z_{t+1} - z_{t+\frac{1}{2}} \right\|^2 - \left\| z_{t+\frac{1}{2}} - z_t \right\|^2$$
$$\leq \|z_t - p\|^2 + 2\eta L \left\| z_{t+\frac{1}{2}} - z_{t-\frac{1}{2}} \right\| \left\| z_{t+\frac{1}{2}} - z_{t+1} \right\| + 2\eta \left\| G_t(z_{t+\frac{1}{2}}) - G_{t-1}(z_{t-\frac{1}{2}}) \right\| \left\| z_{t+\frac{1}{2}} - z_{t+1} \right\|$$
$$- 2\eta\langle F_t(z_{t+\frac{1}{2}}), z_{t+\frac{1}{2}} - p \rangle - \left\| z_{t+1} - z_{t+\frac{1}{2}} \right\|^2 - \left\| z_{t+\frac{1}{2}} - z_t \right\|^2 \tag{31}$$

Since

$$2 \left\| z_{t+\frac{1}{2}} - z_{t-\frac{1}{2}} \right\| \left\| z_{t+\frac{1}{2}} - z_{t+1} \right\|$$
$$\leq \frac{1}{\sqrt{2}} \|z_{t+\frac{1}{2}} - z_{t-\frac{1}{2}}\|^2 + \sqrt{2}\|z_{t+\frac{1}{2}} - z_{t+1}\|^2 \tag{32}$$
$$\leq (1 + \sqrt{2})\|z_{t+\frac{1}{2}} - z_t\|^2 + \|z_t - z_{t-\frac{1}{2}}\|^2 + \sqrt{2}\|z_{t+\frac{1}{2}} - z_{t+1}\|^2$$

we obtain

$$\|z_{t+1} - p\|^2$$

$$\leq \|z_t - p\|^2 - 2\eta\langle F_t(z_{t+\frac{1}{2}}), z_{t+\frac{1}{2}} - p\rangle + ((1+\sqrt{2})\eta L - 1)\|z_{t+\frac{1}{2}} - z_t\|^2$$

$$+ \eta L\|z_t - z_{t-\frac{1}{2}}\|^2 - (1 - \sqrt{2}\eta L)\|z_{t+1} - z_{t+\frac{1}{2}}\|^2 + 2\eta\left\|G_t(z_{t+\frac{1}{2}}) - G_{t-1}(z_{t-\frac{1}{2}})\right\|\left\|z_{t+\frac{1}{2}} - z_{t+1}\right\|$$

(33)

so that

$$\left(1 - (1+\sqrt{2})\eta L\right)\left\|z_{t+\frac{1}{2}} - z_t\right\|^2$$

$$\leq \|z_t - p\|^2 - \|z_{t+1} - p\|^2 - 2\eta\left\langle F(z_{t+\frac{1}{2}}), z_{t+\frac{1}{2}} - p\right\rangle + \eta L\left(\left\|z_t - z_{t-\frac{1}{2}}\right\|^2 - \left\|z_{t+1} - z_{t+\frac{1}{2}}\right\|^2\right)$$

$$+ 2\eta\left\|G_t(z_{t+\frac{1}{2}}) - G_{t-1}(z_{t-\frac{1}{2}})\right\|\left\|z_{t+\frac{1}{2}} - z_{t+1}\right\|$$

(34)

if $\eta < \frac{1}{(\sqrt{2}+1)L}$.

Let $p = z^*$ in the inequality above. Since $F_\infty$ is monotone and $z_{t+1/2} = 2z_t - z_{t-1}$, we have

$$-2\eta\left\langle F_t(z_{t+\frac{1}{2}}), z_{t+\frac{1}{2}} - z^*\right\rangle$$

$$= -2\eta\left\langle F_t(z_{t+\frac{1}{2}}) - F_t(z^*), z_{t+\frac{1}{2}} - z^*\right\rangle - 2\eta\left\langle F_t(z^*), z_{t+\frac{1}{2}} - z^*\right\rangle$$

$$\leq -2\eta\left\langle 2G_t(z_{t+\frac{1}{2}}) - G(z^*), z_{t+\frac{1}{2}} - z^*\right\rangle - 2\eta\left\langle F_\infty(z^*), z_{t+\frac{1}{2}} - z^*\right\rangle$$

$$= -2\eta\left\langle 2G_t(z_{t+\frac{1}{2}}) - G_t(z^*), z_{t+\frac{1}{2}} - z^*\right\rangle + 2\eta\langle F_\infty(z^*), z_{t-1} - z^*\rangle - 4\eta\langle F_\infty(z^*), z_t - z^*\rangle$$

$$\leq -2\eta\left\langle 2G_t(z_{t+\frac{1}{2}}) - G_t(z^*), z_{t+\frac{1}{2}} - z^*\right\rangle + 2\eta\langle F_\infty(z^*), z_{t-1} - z^*\rangle - 2\eta\langle F_\infty(z^*), z_t - z^*\rangle$$

(35)

Also, since $z_t \in \mathcal{Z}$ and $z^*$ is a Nash equilibrium, $\forall t \geq 0$, we have $\langle F_\infty(z^*), z_t - z^*\rangle \geq 0$. By combining (34) and (35), telescoping the terms for $t = 1, 2, \cdots, T$ and dividing both sides by $1 - (1+\sqrt{2})\eta L > 0$, we obtain

$$\sum_{t=1}^{T}\left\|z_{t+\frac{1}{2}} - z_t\right\|^2$$

$$\leq \frac{\|z_1 - z^*\|^2 + \left\|z_1 - z_{\frac{1}{2}}\right\|^2 + 2\eta\langle F_\infty(z^*), z_0 - z^*\rangle}{1 - (1+\sqrt{2})\eta L} + \frac{1}{1 - (1+\sqrt{2})\eta L}$$

$$\cdot \sum_{t=1}^{T}\left(2\eta\left\|G_t(z_{t+\frac{1}{2}}) - G_{t-1}(z_{t-\frac{1}{2}})\right\|\left\|z_{t+\frac{1}{2}} - z_{t+1}\right\| - 2\eta\left\langle 2G_t(z_{t+\frac{1}{2}}) - G_t(z^*), z_{t+\frac{1}{2}} - z^*\right\rangle\right)$$

$$\leq \frac{1}{1 - (1+\sqrt{2})\eta L}\left(\|z_1 - z^*\|^2 + \left\|z_1 - z_{\frac{1}{2}}\right\|^2 + 2\eta\langle F_\infty(z^*), z_0 - z^*\rangle + 8\eta D_{\mathcal{Z}}\sum_{t=1}^{T}\max\|G_t\|\right.$$

$$\left. + 2\eta D_{\mathcal{Z}}\sum_{t=1}^{T}\max\|G_{t-1}\|\right)$$

(36)

With the denotion $H^2 := \left(\|z_1 - z^*\|^2 + \left\|z_1 - z_{\frac{1}{2}}\right\|^2 + 2\eta\langle F_\infty(z^*), z_0 - z^*\rangle + 8\eta D_{\mathcal{Z}}\sum_{t=1}^{T}\max\|G_t\| + 2\eta D_{\mathcal{Z}}\sum_{t=1}^{T}\max\|G_{t-1}\|\right)$, the proof is completed. □

**Lemma 8.** *In the settings of Theorem 2, $\forall T \geq 1$, there exists $C_0 \geq 0$ satisfying*

$$P_{t^*, t\in[[T]]} \leq \frac{C_0}{T} \tag{37}$$

*where $P_{t^*, t\in[[T]]} := \min_{t\in[[T]]} P_t$.*

*Proof.* We first show an upper bound for $P_t$.

$$P_t = \|F_{t-1}(z_t) + c_t\|^2 + \left\|F_{t-1}(z_t) - F_{t-1}(z_{t-\frac{1}{2}})\right\|^2$$

$$= \left\|F_{t-1}(z_t) - F_{t-1}(z_{t-\frac{1}{2}}) + \frac{z_{t-\frac{1}{2}} - z_{t-1}}{\eta}\right\|^2 + \left\|F(z_t) - F(z_{t-\frac{1}{2}})\right\|^2$$

$$\leq 3\left\|F_{t-1}(z_t) - F_{t-1}(z_{t-\frac{1}{2}})\right\|^2 + \frac{2}{\eta^2}\|z_t - z_{t-1}\|^2$$

$$= 3\left\|F_\infty(z_t) - F_\infty(z_{t-\frac{1}{2}})\right\|^2 + 6\left\langle F_\infty(z_t) - F_\infty(z_{t-\frac{1}{2}}), G_{t-1}(z_t) - G_{t-1}(z_{t-\frac{1}{2}})\right\rangle$$

$$\quad + 3\left\|G_{t-1}(z_t) - G_{t-1}(z_{t-\frac{1}{2}})\right\|^2 + \frac{2}{\eta^2}\|z_t - z_{t-1}\|^2$$

$$\leq 3L^2\left\|z_t - z_{t-\frac{1}{2}}\right\|^2 + \frac{2}{\eta^2}\left\|z_{t-1} - z_{t-\frac{1}{2}}\right\|^2$$

$$\quad + 6\left\|F_\infty(z_t) - F_\infty(z_{t-\frac{1}{2}})\right\|\left\|G_{t-1}(z_t) - G_{t-1}(z_{t-\frac{1}{2}})\right\| + 3\left\|G_{t-1}(z_t) - G_{t-1}(z_{t-\frac{1}{2}})\right\|^2$$

$$\leq 3L^2\left\|z_t - z_{t-1} + z_{t-1} - z_{t-\frac{1}{2}}\right\|^2 + \frac{2}{\eta^2}\|z_t - z_{t-1}\|^2 + 6L\left\|z_t - z_{t-\frac{1}{2}}\right\|\left\|G_{t-1}(z_t) - G_{t-1}(z_{t-\frac{1}{2}})\right\|$$

$$\quad + 3\left\|G_{t-1}(z_t) - G_{t-1}(z_{t-\frac{1}{2}})\right\|^2$$

$$\leq 6L^2\left\|z_{t-\frac{1}{2}} - z_{t-1}\right\|^2 + \left(\frac{2}{\eta^2} + 6L^2\right)\|z_t - z_{t-1}\|^2 + 12LD_{\mathcal{Z}}\max\|G_{t-1}\| + 12\max\|G_{t-1}\|^2$$

$$\leq \frac{2 + 6\eta^2 L^2}{\eta^2}\left(\left\|z_{t-\frac{1}{2}} - z_{t-1}\right\|^2 + \|z_t - z_{t-1}\|^2\right) + 12LD_{\mathcal{Z}}\max\|G_{t-1}\| + 12\max\|G_{t-1}\|^2$$

$$\tag{38}$$

Summing the above inequality of $t = 1, 2, \cdots T$, we get

$$\sum_{t=1}^{T} P_t$$

$$\leq \frac{2 + 6\eta^2 L^2}{\eta^2}\sum_{t=1}^{T}\left(\left\|z_{t-\frac{1}{2}} - z_{t-1}\right\|^2 + \|z_t - z_{t-1}\|^2\right) + 12LD_{\mathcal{Z}}\sum_{t=1}^{T}\max\|G_{t-1}\| + 12\sum_{t=1}^{T}\max\|G_{t-1}\|^2$$

$$= \frac{2 + 6\eta^2 L^2}{\eta^2}\left(\|z_1 - z_0\|^2 + \sum_{t=1}^{T-1}\left(\left\|z_{t+\frac{1}{2}} - z_t\right\|^2 + \|z_{t+1} - z_t\|^2\right)\right) + 12LD_{\mathcal{Z}}\sum_{t=1}^{T}\max\|G_{t-1}\|$$

$$\quad + 12\sum_{t=1}^{T}\max\|G_{t-1}\|^2$$

$$\leq \frac{2 + 6\eta^2 L^2}{\eta^2}\left(\|z_1 - z_0\|^2 + \frac{2H^2}{1 - (1+\sqrt{2})\eta L}\right) + 12LD_{\mathcal{Z}}\sum_{t=1}^{T}\max\|G_{t-1}\| + 12\sum_{t=1}^{T}\max\|G_{t-1}\|^2$$

$$\leq \frac{6(1 + 3\eta^2 L^2)H^2}{\eta^2(1 - (1+\sqrt{2})\eta L)} + 12LD_{\mathcal{Z}}\sum_{t=1}^{\infty}\max\|G_{t-1}\| + 12\sum_{t=1}^{\infty}\max\|G_{t-1}\|^2$$

$$\tag{39}$$

The second last inequality holds by Lemma 7. The last inequality holds since $\|z_1 - z_0\|^2 \leq \frac{4}{L^2}\|F(z_0)\|^2 \leq H^2$. Denote RHS of the last inequality in (39) as $C_0$, and we have

$$P_{t^*, t\in[[T]]} \leq \frac{C_0}{T} \tag{40}$$

This completes the proof. $\qquad\square$

### D.3 PROOF OF LEMMA 2

If $G_t$ is $L_{G_t}$-Lipschitz and $G_t(z^*) = 0$, we have

$$-2\eta\Big\langle F_t(z_{t+\frac{1}{2}}), z_{t+\frac{1}{2}} - z^*\Big\rangle$$

$$\leq -2\eta\Big\langle 2G_t(z_{t+\frac{1}{2}}) - G_t(z^*), z_{t+\frac{1}{2}} - z^*\Big\rangle + 2\eta\langle F_\infty(z^*), z_{t-1} - z^*\rangle - 2\eta\langle F_\infty(z^*), z_t - z^*\rangle$$

$$= -2\eta\Big\langle 2G_t(z_{t+\frac{1}{2}}) - 2G_t(z^*), z_{t+\frac{1}{2}} - z^*\Big\rangle + 2\eta\langle F_\infty(z^*), z_{t-1} - z^*\rangle - 2\eta\langle F_\infty(z^*), z_t - z^*\rangle$$

$$= -4\eta\Big\langle G_t(z_{t+\frac{1}{2}}) - G_t(z_t), z_{t+\frac{1}{2}} - z_t\Big\rangle - 4\eta\Big\langle G_t(z_t) - G_t(z^*), z_t - z^*\Big\rangle$$

$$+ 2\eta\langle F_\infty(z^*), z_{t-1} - z^*\rangle - 2\eta\langle F_\infty(z^*), z_t - z^*\rangle$$

$$\leq 4\eta L_{G_t}\left\|z_{t+\frac{1}{2}} - z_t\right\|^2 + 4\eta L_{G_t}\|z_t - z^*\|^2 + 2\eta\langle F_\infty(z^*), z_{t-1} - z^*\rangle - \frac{2\eta}{1 + 4\eta L_{G_t}}\langle F_\infty(z^*), z_t - z^*\rangle$$

$$(41)$$

so that

$$\left(1 - (1 + \sqrt{2})\eta L - 4\eta L_{G_t}\right)\left\|z_{t+\frac{1}{2}} - z_t\right\|^2$$

$$\leq (1 + 4\eta L_{G_t})\|z_t - z^*\|^2 - \|z_{t+1} - z^*\|^2 + 2\eta\langle F_\infty(z^*), z_{t-1} - z^*\rangle - \frac{2\eta}{1 + 4\eta L_{G_t}}\langle F_\infty(z^*), z_t - z^*\rangle$$

$$+ \eta L\left(\left\|z_t - z_{t-\frac{1}{2}}\right\|^2 - \left\|z_{t+1} - z_{t+\frac{1}{2}}\right\|^2\right) + 2\eta\Big\langle G_t(z_{t+\frac{1}{2}}) - G_{t-1}(z_{t-\frac{1}{2}}), z_{t+\frac{1}{2}} - z_{t+1}\Big\rangle$$

$$\leq (1 + 4\eta L_{G_t})\|z_t - z^*\|^2 - \|z_{t+1} - z^*\|^2 + 2\eta\langle F_\infty(z^*), z_{t-1} - z^*\rangle - \frac{2\eta}{1 + 4\eta L_{G_t}}\langle F_\infty(z^*), z_t - z^*\rangle$$

$$+ \eta L\left(\left\|z_t - z_{t-\frac{1}{2}}\right\|^2 - \left\|z_{t+1} - z_{t+\frac{1}{2}}\right\|^2\right) + 2\eta\Big\langle G_t(z_{t+\frac{1}{2}}) - G_t(z^*), z_{t+\frac{1}{2}} - z_{t+1}\Big\rangle$$

$$+ 2\eta\Big\langle G_{t-1}(z^*) - G_{t-1}(z_{t-\frac{1}{2}}), z_{t+\frac{1}{2}} - z_{t+1}\Big\rangle$$

$$\leq (1 + 4\eta L_{G_t})\|z_t - z^*\|^2 - \|z_{t+1} - z^*\|^2 + 2\eta\langle F_\infty(z^*), z_{t-1} - z^*\rangle - \frac{2\eta}{1 + 4\eta L_{G_t}}\langle F_\infty(z^*), z_t - z^*\rangle$$

$$+ \eta L\left(\left\|z_t - z_{t-\frac{1}{2}}\right\|^2 - \frac{1}{1 + 4\eta L_{G_t}}\left\|z_{t+1} - z_{t+\frac{1}{2}}\right\|^2\right) + 2\eta\left\|G_t(z_{t+\frac{1}{2}}) - G_{t-1}(z_{t-\frac{1}{2}})\right\|\left\|z_{t+\frac{1}{2}} - z_{t+1}\right\|$$

$$(42)$$

Since

$$2\eta\left\|G_t(z_{t+\frac{1}{2}}) - G_{t-1}(z_{t-\frac{1}{2}})\right\|\left\|z_{t+\frac{1}{2}} - z_{t+1}\right\|$$

$$\leq 2\eta\left\|G_t(z_{t+\frac{1}{2}}) - G_t(z^*)\right\|\left\|z_{t+\frac{1}{2}} - z_{t+1}\right\| + 2\eta\left\|G_{t-1}(z^*) - G_{t-1}(z_{t-\frac{1}{2}})\right\|\left\|z_{t+\frac{1}{2}} - z_{t+1}\right\|$$

$$\leq 2\eta L_{G_t}\left\|z_{t+\frac{1}{2}} - z^*\right\|\left\|z_{t+\frac{1}{2}} - z_{t+1}\right\| + 2\eta L_{G_{t-1}}\left\|z^* - z_{t-\frac{1}{2}}\right\|\left\|z_{t+\frac{1}{2}} - z_{t+1}\right\|$$

$$\leq 2\eta L_{G_t}\|z_{t+1} - z^*\|\left\|z_{t+\frac{1}{2}} - z_{t+1}\right\| + 2\eta L_{G_t}\left\|z_{t+\frac{1}{2}} - z_{t+1}\right\|^2 + 2\eta L_{G_{t-1}}\left\|z^* - z_{t-\frac{1}{2}}\right\|\left\|z_{t+\frac{1}{2}} - z_{t+1}\right\|$$

$$\leq 2\eta L_{G_t}\|z_{t+1} - z^*\|\left\|z_{t+\frac{1}{2}} - z_{t+1}\right\| + 2\eta L_{G_t}\left\|z_{t+\frac{1}{2}} - z_{t+1}\right\|^2 + 2\eta L_{G_{t-1}}\|z^* - z_t\|\left\|z_{t+\frac{1}{2}} - z_{t+1}\right\|$$

$$+ 2\eta L_{G_{t-1}}\left\|z_t - z_{t-\frac{1}{2}}\right\|\left\|z_{t+\frac{1}{2}} - z_{t+1}\right\|$$

$$\leq \eta L_{G_t}\|z_{t+1} - z^*\|^2 + (3\eta L_{G_t} + 2\eta L_{G_{t-1}})\left\|z_{t+\frac{1}{2}} - z_{t+1}\right\|^2 + \eta L_{G_{t-1}}\|z^* - z_t\|^2 + \eta L_{G_{t-1}}\left\|z_t - z_{t-\frac{1}{2}}\right\|^2$$

$$(43)$$

we have

$$\left(1 - (1 + \sqrt{2})\eta L - 4\eta L_{G_t}\right)\left\|z_{t+\frac{1}{2}} - z_t\right\|^2$$

$$\leq (1 + 4\eta L_{G_t})\|z_t - z^*\|^2 - \|z_{t+1} - z^*\|^2 + 2\eta\langle F_\infty(z^*), z_{t-1} - z^*\rangle - \frac{2\eta}{1 + 4\eta L_{G_t}}\langle F_\infty(z^*), z_t - z^*\rangle$$

$$+ \eta L \left( \left\| z_t - z_{t-\frac{1}{2}} \right\|^2 - \frac{1}{1 + 4\eta L_{G_t}} \left\| z_{t+1} - z_{t+\frac{1}{2}} \right\|^2 \right) + 2\eta \left\| G_t(z_{t+\frac{1}{2}}) - G_{t-1}(z_{t-\frac{1}{2}}) \right\| \left\| z_{t+\frac{1}{2}} - z_{t+1} \right\|$$

$$\leq (1 + 4\eta L_{G_t}) \|z_t - z^*\|^2 - \|z_{t+1} - z^*\|^2 + 2\eta\langle F_\infty(z^*), z_{t-1} - z^* \rangle - \frac{2\eta}{1 + 4\eta L_{G_t}}\langle F_\infty(z^*), z_t - z^* \rangle$$

$$+ \eta L \left( \left\| z_t - z_{t-\frac{1}{2}} \right\|^2 - \frac{1}{1 + 4\eta L_{G_t}} \left\| z_{t+1} - z_{t+\frac{1}{2}} \right\|^2 \right) + \eta L_{G_t} \|z_{t+1} - z^*\|^2 + (3\eta L_{G_t} + 2\eta L_{G_{t-1}}) \left\| z_{t+\frac{1}{2}} - z_{t+1} \right\|^2$$

$$+ \eta L_{G_{t-1}} \|z^* - z_t\|^2 + \eta L_{G_{t-1}} \left\| z_t - z_{t-\frac{1}{2}} \right\|^2$$

$$= \left( 1 + 4\eta L_{G_t} + \eta L_{G_{t-1}} \right) \|z_t - z^*\|^2 - (1 - \eta L_{G_t}) \|z_{t+1} - z^*\|^2 + 2\eta\langle F_\infty(z^*), z_{t-1} - z^* \rangle$$

$$- \frac{2\eta}{1 + 4\eta L_{G_t}}\langle F_\infty(z^*), z_t - z^* \rangle + \eta \left( L + L_{G_{t-1}} \right) \left\| z_t - z_{t-\frac{1}{2}} \right\|^2$$

$$- \left( \frac{\eta L}{1 + 4\eta L_{G_t}} - 3\eta L_{G_t} - 2\eta L_{G_{t-1}} \right) \left\| z_{t+1} - z_{t+\frac{1}{2}} \right\|^2 \tag{44}$$

Since

$$\frac{1 + 4\eta L_{G_t} + \eta L_{G_{t-1}}}{1 - \eta L_{G_t}} - \frac{\eta(L + L_{G_{t-1}})}{\frac{\eta L}{1+4\eta L_{G_t}} - 3\eta L_{G_t} - 2\eta L_{G_{t-1}}}$$

$$= ((\eta L - (1 + 4\eta L_{G_t})(3\eta L_{G_t} + 2\eta L_{G_{t-1}}))(4\eta L_{G_t} + \eta L_{G_{t-1}} + 1) - \eta(1 - \eta L_{G_t})(1 + 4\eta L_{G_t})(L + L_{G_{t-1}}))$$

$$(1 - \eta L_{G_t})^{-1} \left( \eta L \left( 1 + 4\eta L_{G_t} \right) \left( -3\eta L_{G_t} - 2\eta L_{G_{t-1}} \right) \right)^{-1}$$

$$= (-48\eta^3 L_{G_t}^3 - 40\eta^3 L_{G_t}^2 L_{G_{t-1}} + (4L\eta^3 - 24\eta^2)L_{G_t}^2 - 8\eta^3 L_{G_t} L_{G_{t-1}}^2 - 22\eta^2 L_{G_t} L_{G_{t-1}} + (L\eta^2 - 3\eta)L_{G_t}$$

$$- 2\eta^2 L_{G_{t-1}}^2 + (L\eta^2 - 3\eta)L_{G_{t-1}}) (1 - \eta L_{G_t})^{-1} \left( \eta L \left( 1 + 4\eta L_{G_t} \right) \left( -3\eta L_{G_t} - 2\eta L_{G_{t-1}} \right) \right)^{-1} \tag{45}$$

and $\frac{1 + 4\eta L_{G_t} + \eta L_{G_{t-1}}}{1 - \eta L_{G_t}} - (1 + 4\eta L_{G_t}) = \frac{4L_{G_t}^2 \eta^2 + L_{G_t}\eta + L_{G_{t-1}}\eta}{1 - \eta L_{G_t}}$, there exists $T_N$ and $\epsilon > 0$ so that $\forall t > T_N$, it holds that $L_{G_t}, L_{G_{t-1}} < \epsilon$, $1 - (1 + \sqrt{2})\eta L - 4\eta L_{G_t} > 0$, $\frac{1 + 4\eta L_{G_t} + \eta L_{G_{t-1}}}{1 - \eta L_{G_t}} < \frac{\eta(L + L_{G_{t-1}})}{\frac{\eta L}{1+4\eta L_{G_t}} - 3\eta L_{G_t} - 2\eta L_{G_{t-1}}}$ with $L\eta^2 - 3\eta < 0$ and $\frac{1 + 4\eta L_{G_t} + \eta L_{G_{t-1}}}{1 - \eta L_{G_t}} > 1 + 4\eta L_{G_t}$. Since $\frac{\eta(L + L_{G_{t-1}})}{\frac{\eta L}{1+4\eta L_{G_t}} - 3\eta L_{G_t} - 2\eta L_{G_{t-1}}} = 1 + \frac{4\eta L + 6}{L}\epsilon + O(\epsilon^2)$, we have

$$\sum_{t=T_N+1}^{\infty} \left( \frac{4\eta L + 6}{L} L_{G_{t-1}} + O(L_{G_{t-1}}^2) \right) \leq \sum_{t=T_N+1}^{\infty} \left( \frac{4\eta L + 6}{L} L_{G_{t-1}} + O(L_{G_{t-1}}^2) \right) < \infty \tag{46}$$

Hence,

$$\prod_{t=T_N+1}^{\infty} \frac{\eta(L + L_{G_{t-1}})}{\frac{\eta L}{1+4\eta L_{G_t}} - 3\eta L_{G_t} - 2\eta L_{G_{t-1}}} < \infty \tag{47}$$

so that

$$E := \prod_{t=2}^{\infty} \frac{\eta(L + L_{G_{t-1}})}{\frac{\eta L}{1+4\eta L_{G_t}} - 3\eta L_{G_t} - 2\eta L_{G_{t-1}}} < \infty \tag{48}$$

Hence, $\forall t > T_N$,

$$\|z_{t+1} - z^*\|^2$$

$$\leq \|z_{t+1} - z^*\|^2 + \frac{2\eta}{(1 + 4\eta L_{G_t})(1 - \eta L_{G_t})}\langle F_\infty(z^*), z_t - z^* \rangle$$

$$+ \frac{\eta L + (3\eta L_{G_t} + 2\eta L_{G_{t-1}})(1 + 4\eta L_{G_t})}{(1 + 4\eta L_{G_t})(1 - \eta L_{G_t})} \left\| z_{t+1} - z_{t+\frac{1}{2}} \right\|^2$$

$$\leq E \|z_2 - z^*\|^2 + \frac{2E\eta}{(1 + 4\eta L_{G_1})(1 - \eta L_{G_1})}\langle F_\infty(z^*), z_1 - z^* \rangle$$

$$+ \frac{E\eta L + E(3\eta L_{G_1} + 2\eta L_{G_0})(1 + 4\eta L_{G_1})}{(1 + 4\eta L_{G_1})(1 - \eta L_{G_1})} \left\| z_2 - z_{\frac{3}{2}} \right\|^2 \tag{49}$$

Define $C_1 = \max\Big\{ E\|z_2 - z^*\|^2 + \frac{2E\eta}{(1+4\eta L_{G_1})(1-\eta L_{G_1})}\langle F_\infty(z^*), z_1 - z^*\rangle +$
$\frac{E\eta L + E(3\eta L_{G_1} + 2\eta L_{G_0})(1+4\eta L_{G_1})}{(1+4\eta L_{G_1})(1-\eta L_{G_1})} \cdot \left\|z_2 - z_{\frac{3}{2}}\right\|^2, \max_{t=0,1,2,\cdots,T_N}\|z_T - z^*\|^2 \Big\}$. We have that
$\|z_t - z^*\| < \sqrt{C_1}$, $t \in \mathbb{N}$. Hence, there exists $\mathcal{Z}_L$ satisfying $z^*, z_t \in \mathcal{Z}_L$, $t \in \mathbb{N}$ and we can take $D_{\mathcal{Z}} = D_{\mathcal{Z}_L} = 2\sqrt{C_1}$.

### D.4 FINAL PROOF

With the definition of $P_T$, we have

$$r^{tan}(z_T)^2$$
$$\leq \|F_{T-1}(z_T) - G_{T-1}(z_T) + c_T\|^2$$
$$\leq \|F_\infty(z_T) + c_T\|^2$$
$$\leq \|F_{T-1}(z_T) + c_T\|^2 + 2\langle F_{T-1}(z_T) + c_T, -G_{T-1}(z_T)\rangle + \| -G_{T-1}(z_T)\|^2$$
$$\leq \|F_{T-1}(z_T) + c_T\|^2 + 2\|F_{T-1}(z_T) + c_T\|\|G_{T-1}(z_T)\| + \|G_{T-1}(z_T)\|^2 + \left\|F_{T-1}(z_T) - F_{T-1}(z_{T-\frac{1}{2}})\right\|$$
$$\leq P_T + 2\|F_{T-1}(z_T)\|\max\|G_{T-1}\| + \max\|G_{T-1}\|^2$$

$$(50)$$

Under the BAP assumption, $\max\|G_{T-1}\| = O\left(\frac{1}{T}\right)$. Hence, according to Lemma 6, we have

$$P_{\lceil\frac{T}{2}\rceil} \leq P_1 + \left(\frac{2}{\eta} + 8L\right)D_{\mathcal{Z}}\sum_{t=1}^{\lceil\frac{T}{2}\rceil - 1}\max\|G_t\| + \frac{2}{\eta}D_{\mathcal{Z}}\sum_{t=1}^{\lceil\frac{T}{2}\rceil - 1}\max\|G_{t-1}\| + 8\sum_{t=1}^{\lceil\frac{T}{2}\rceil - 1}\max\|G_t\|^2$$

$$\leq P_1 + \left(\frac{2}{\eta} + 8L\right)D_{\mathcal{Z}}\sum_{t=1}^{\lceil\frac{T}{2}\rceil - 1}\max\|G_t\| + \frac{2}{\eta}D_{\mathcal{Z}}\sum_{t=1}^{\infty}\max\|G_{t-1}\| + 8\sum_{t=1}^{\infty}\max\|G_t\|^2 =: P_{hs}$$

$$(51)$$

$$P_T \leq P_{t^*, t\in[\lceil\frac{T}{2}\rceil, T]} + \left(\frac{2}{\eta} + 8L\right)D_{\mathcal{Z}}\sum_{t=\lceil\frac{T}{2}\rceil - 1}^{T}\max\|G_t\| + \frac{2}{\eta}D_{\mathcal{Z}}\sum_{t=\lceil\frac{T}{2}\rceil - 1}^{T}\max\|G_{t-1}\| + 8\sum_{t=\lceil\frac{T}{2}\rceil - 1}^{T}\max\|G_t\|^2$$

$$\leq P_{t^*, t\in[\lceil\frac{T}{2}\rceil, T]} + \left(\frac{2}{\eta} + 8L\right)D_{\mathcal{Z}}\sum_{t=\lceil\frac{T}{2}\rceil - 1}^{\infty}\max\|G_t\| + \frac{2}{\eta}D_{\mathcal{Z}}\sum_{t=\lceil\frac{T}{2}\rceil - 1}^{\infty}\max\|G_{t-1}\| + 8\sum_{t=\lceil\frac{T}{2}\rceil - 1}^{\infty}\max\|G_t\|^2$$

$$(52)$$

Hence,

$$r^{tan}(z_T)^2$$
$$\leq P_{t^*, t\in[\lceil\frac{T}{2}\rceil, T]} + \left(\frac{2}{\eta} + 8L\right)D_{\mathcal{Z}}\sum_{t=t^*, t\in[\lceil\frac{T}{2}\rceil, T]}^{T}\max\|G_t\| + \frac{2}{\eta}D_{\mathcal{Z}}\sum_{t=t^*, t\in[\lceil\frac{T}{2}\rceil, T]}^{T}\max\|G_{t-1}\|$$
$$+ 8\sum_{t=t^*, t\in[\lceil\frac{T}{2}\rceil, T]}^{T}\max\|G_t\|^2 + 2\|F_{T-1}(z_T)\|\max\|G_{T-1}\| + \max\|G_{T-1}\|^2$$
$$\leq P_{t^*, t\in[\lceil\frac{T}{2}\rceil, T]} + \left(\frac{2}{\eta} + 8L\right)D_{\mathcal{Z}}\sum_{t=\lceil\frac{T}{2}\rceil - 1}^{\infty}\max\|G_t\| + \frac{2}{\eta}D_{\mathcal{Z}}\sum_{t=\lceil\frac{T}{2}\rceil - 1}^{\infty}\max\|G_{t-1}\| + 8\sum_{t=\lceil\frac{T}{2}\rceil - 1}^{\infty}\max\|G_t\|^2$$
$$+ 2\|F_{T-1}(z_T)\|\max\|G_{T-1}\| + \max\|G_{T-1}\|^2$$

$$(53)$$

If $\sum_{t=T}^{\infty}\max\|G_t\| = O\left(\sum_{t=\lceil\frac{T}{2}\rceil - 1}^{\infty}\max\|G_t\|\right)$, $\sum_{t=\lceil\frac{T}{2}\rceil - 1}^{\infty}\max\|G_t\| = O\left(\frac{1}{T}\right)$. Since $\max\|G_{T-1}\|^2 = O\left(\sum_{t=\lceil\frac{T}{2}\rceil - 1}^{\infty}\max\|G_t\|^2\right)$, $\max\|G_{T-1}\| = O\left(\sum_{t=\lceil\frac{T}{2}\rceil - 1}^{\infty}\max\|G_t\|\right)$,

$\sum_{t=\lceil \frac{T}{2} \rceil - 1}^{\infty} \max \|G_t\|^2 = O\left( \sum_{t=\lceil \frac{T}{2} \rceil - 1}^{\infty} \max \|G_t\| \right)$ and $P_{t^*, t \in \left[\lceil \frac{T}{2} \rceil, T\right]} = O\left(\frac{1}{T}\right)$, we have

$$r^{tan}(z_T)^2 = \max \left\{ O\left(\frac{1}{T}\right), O\left( \sum_{t=T/2}^{\infty} \max \|G_t\| \right) \right\} \tag{54}$$

Hence,

$$r^{tan}(z_T) = \max \left\{ O\left(\frac{1}{\sqrt{T}}\right), O\left( \sqrt{\sum_{t=T/2}^{\infty} \max \|G_t\|} \right) \right\} \tag{55}$$

## E    MISSING PROOFS OF THEOREM 3

In periodic games, each period as a whole is time-invariant (Franke & Selgrade (2003), Section 3). We take advantage of this property in our analysis.

With the ARG algorithm, we have

$$\begin{bmatrix} x_0 \\ y_0 \\ x_t \\ y_t \\ x_{t+1} \\ y_{t+1} \end{bmatrix} = \begin{bmatrix} I_3 & O_{3\times 3} & O_{3\times 3} \\ O_{3\times 3} & O_{3\times 3} & I_3 \\ Q_t & R_t & S_t \end{bmatrix} \begin{bmatrix} x_0 \\ y_0 \\ x_{t-1} \\ y_{t-1} \\ x_t \\ y_t \end{bmatrix} \tag{56}$$

where

$$Q_t = \begin{bmatrix} \frac{1}{t+1} I & \frac{\eta A_t}{t(t+1)} \\ -\frac{\eta A_t^T}{t(t+1)} & \frac{1}{t+1} I \end{bmatrix} \tag{57}$$

$$R_t = \begin{bmatrix} 0 & \eta \left(1 - \frac{1}{t}\right) A_t \\ -\eta \left(1 - \frac{1}{t}\right) A_t^T & 0 \end{bmatrix} \tag{58}$$

$$S_t = \begin{bmatrix} \left(1 - \frac{1}{t+1}\right) I & -\eta \left(2 - \frac{1}{t+1}\right) A_t \\ \eta \left(2 - \frac{1}{t+1}\right) A_t^T & \left(1 - \frac{1}{t+1}\right) I \end{bmatrix} \tag{59}$$

Denote $\begin{bmatrix} I_3 & O_{3\times 3} & O_{3\times 3} \\ O_{3\times 3} & O_{3\times 3} & I_3 \\ Q_t & R_t & S_t \end{bmatrix}$ as $P_t$. We have

$$\lambda I - \lim_{t \to \infty} P_{t+1} P_t = \lambda I - \begin{bmatrix} I_3 & O_{3\times 3} & O_{3\times 3} \\ O_{3\times 3} & O_{3\times 3} & I_3 \\ O_{3\times 3} & R_t' & S_t' \end{bmatrix} \tag{60}$$

where

$$R_t' = \begin{bmatrix} 0 & \eta A_t \\ -\eta A_t^T & 0 \end{bmatrix} = \begin{cases} \begin{bmatrix} 0 & \eta & -\eta \\ -\eta & 0 & 0 \\ \eta & 0 & 0 \end{bmatrix}, t \text{ is odd} \\ \begin{bmatrix} 0 & -\eta & \eta \\ \eta & 0 & 0 \\ -\eta & 0 & 0 \end{bmatrix}, t \text{ is even} \end{cases} \tag{61}$$

$$S_t' = \begin{bmatrix} I & -2\eta A_t \\ 2\eta A_t^T & I \end{bmatrix} = \begin{cases} \begin{bmatrix} 1 & -2\eta & 2\eta \\ 2\eta & 1 & 0 \\ -2\eta & 0 & 1 \end{bmatrix}, t \text{ is odd} \\ \begin{bmatrix} 1 & 2\eta & -2\eta \\ -2\eta & 1 & 0 \\ 2\eta & 0 & 1 \end{bmatrix}, t \text{ is even} \end{cases} \tag{62}$$

Hence,

$$\det\left(\lambda I - \lim_{t\to\infty} P_{t+1}P_t\right) = \lambda(\lambda-1)^4(8\eta^2\lambda - 2\eta^2 - \lambda^2 + \lambda)^2 \tag{63}$$

which means that $\frac{\sqrt{64\eta^4+8\eta^2+1}+8\eta^2+1}{2}$ is an eigenvalue of $\lim_{t\to\infty} P_{t+1}P_t$ and it is greater than 1.

Since an eigenvalue of $\lim_{t\to\infty} P_{t+1}P_t$ is $\frac{\sqrt{64\eta^4+8\eta^2+1}+8\eta^2+1}{2} > 1$, denote this eigenvalue as $\rho_\infty$ and the corresponding eigenvector of $\lim_{t\to\infty} P_{t+1}P_t$ as $v$. Then, we obtain

$$\left(\lim_{t\to\infty}\prod_{i=0}^{k} P_{t+2i+1}P_{t+2i}\right) v = \left(\prod_{i=0}^{k}\lim_{t\to\infty} P_{t+1}P_t\right) v = \rho_\infty^{k+1}v \tag{64}$$

which means that $\forall \epsilon > 0$, there exists $M > 0$ so that $\forall t > M$, there exists

$$\left\|\left(\prod_{i=0}^{k} P_{t+2i+1}P_{t+2i}\right) v\right\| - \left\|\rho_\infty^{k+1}v\right\| < \epsilon \tag{65}$$

Since the action set is $\mathbb{R}^3$, we have

$$r^{tan}(z) = F(z) = \begin{bmatrix} Ay \\ -A^T x \end{bmatrix} \tag{66}$$

Hence,

$$\left\|r^{tan}(z)\right\| = \left\|\begin{bmatrix} Ay \\ -A^T x \end{bmatrix}\right\| = \|z\| \tag{67}$$

Since $\left\|\rho_\infty^k v\right\| = \|\rho_\infty\|^k \|v\|$ grows exponentially, $\|z_{t+2k}\|$ does not converge. With the ARG algorithm, $\forall v$, corresponding $z_0$ and $z_{1/2}$ exist for $z_t = v$. This means that the tangent residual with the ARG algorithm diverges exponentially if such $z_0$ and $z_{1/2}$ are the initial actions. This completes the proof.

# F   MISSING PROOFS OF THEOREM 4

To prove Theorem 4, we apply a potential function argument. We first show the potential function is approximately non-increasing and then prove that it is upper bounded by a term independent of $T$. As the potential function at step $t$ is also at least $\Omega(t^2)r^{tan}(z_t)^2$, we conclude that the ARG algorithm has a $O(\frac{1}{T})$ convergence rate.

## F.1   POTENTIAL FUNCTION

With the update rules of the ARG algorithm, i.e., $z_0, z_{1/2} \in \mathbb{R}^n$ being initial points, for $t \geq 1$,

$$\begin{aligned} z_{t+\frac{1}{2}} &= 2z_t - z_{t-1} + \frac{1}{t+1}(z_0 - z_t) - \frac{1}{t}(z_0 - z_{t-1}) \\ z_{t+1} &= \Pi_{\mathcal{Z}}\left[z_t - \eta F_t(z_{t+\frac{1}{2}}) + \frac{1}{t+1}(z_0 - z_t)\right] \end{aligned} \tag{68}$$

Since $N_{\mathcal{Z}}$ is the normal cone of a closed convex set $\mathcal{Z}$ and $\Pi_{\mathcal{Z}}$ is the projection to set $\mathcal{Z}$, if we apply the ARG algorithm to solve time-varying game problems, the algorithm calculates gradient function $F_t(z)$ once and a projection to $\mathcal{Z}$ once per iteration. Next, we specify the potential function. Define

$$c_{t+1} := \frac{z_t - \eta F_t(z_{t+\frac{1}{2}}) + \frac{1}{t+1}(z_0 - z_t) - z_{t+1}}{\eta}, \quad \forall t \geq 0 \tag{69}$$

By update rule we have $c_t \in N_{\mathcal{Z}}$ for all $t \geq 1$. The potential function at $t \geq 1$ is defined as

$$V_t := \frac{t(t+1)}{2}\|\eta F_{t-1}(z_t) + \eta c_t\|^2 + t\langle \eta F_{t-1}(z_t) + \eta c_t, z_t - z_0\rangle + \frac{t(t+1)}{2}\left\|\eta F_{t-1}(z_t) - \eta F_{t-1}(z_{t-\frac{1}{2}})\right\|^2 \tag{70}$$

### F.2 APPROXIMATE MONOTONICITY OF THE POTENTIAL FUNCTION

**Lemma 9.** $\forall L > 0$ and $\rho \geq -\frac{1}{60L}$, There exists $\eta > 0$ such that

$$\frac{1}{2} - (12 - \frac{4\rho}{\eta})\eta^2 L^2 + \frac{2\rho}{\eta} \geq 0 \tag{71}$$

*Moreover, every $\eta > 0$ satisfies (71) also satisfies $\frac{\rho}{\eta} \geq -\frac{1}{4}$.*

*Proof.* Equation (71) means that

$$\rho > \frac{\eta L(24\eta^2 L^2 - 1)}{4 + 8\eta^2 L^2} \cdot \frac{1}{L} \tag{72}$$

Let $x = \eta L$ and $h(x) = \frac{x(24x^2 - 1)}{4 + 8x^2}$. Since $h(\frac{1}{12}) = -\frac{5}{292} < -\frac{1}{60}$, there exists $\eta = \frac{1}{12L}$ satisfying (71).

Besides, with $\eta L > 0$ and (71), we obtain

$$\frac{\rho}{\eta} \geq -\frac{1 - 72\eta^2 L^2}{4 + 8\eta^2 L^2} \geq -\frac{1}{4} \tag{73}$$

$\square$

We show in the following lemma that $V_t$ is approximately non-increasing.

**Lemma 10.** *With the settings in Theorem 4, $\forall t \geq 1$, we have*

$$\begin{aligned}
V_t &- V_{t+1} \\
&\leq -\frac{1}{8}\|\eta F_t(z_{t+1}) + \eta c_{t+1}\|^2 + t(t+1)\langle \eta G_t(z_{t+1}) - \eta G_{t-1}(z_t), z_{t+1} - z_t\rangle \\
&+ \frac{t(t+1)}{4p}\left(1 - \frac{\rho}{3\eta}\right)\left\|-\eta G_t(z_{t+1}) + \eta G_t(z_{t+\frac{1}{2}})\right\|^2 \\
&+ \frac{t(t+1)}{2p}\left(1 - \frac{\rho}{3\eta}\right)\left\langle \eta F_t(z_{t+1}) - \eta F_t(z_{t+\frac{1}{2}}), -\eta G_t(z_{t+1}) + \eta G_t(z_{t+\frac{1}{2}})\right\rangle
\end{aligned} \tag{74}$$

*Proof.* We show that $V_t - V_{t+1}$ plus a few non-positive terms is still $\geq -\frac{1}{8}\|\eta F(z_{t+1}) + \eta c_{t+1}\|^2 + t(t+1)\langle \eta G_t(z_{t+1})$

$- \quad \eta G_{t-1}(z_t), z_{t+1} \quad - \quad z_t\rangle \quad + \quad \frac{t(t+1)}{4p}\left(1 - \frac{\rho}{3\eta}\right)\left\|-\eta G_t(z_{t+1}) + \eta G_t(z_{t+\frac{1}{2}})\right\|^2 \quad +$

$\frac{t(t+1)}{2p}\left(1 - \frac{\rho}{3\eta}\right)\left\langle \eta F_t(z_{t+1}) - \eta F_t(z_{t+\frac{1}{2}}),$

$-\eta G_t(z_{t+1}) + \eta G_t(z_{t+\frac{1}{2}})\right\rangle$, where $p > 0$.

With the settings of $F_\infty$ in Theorem 4, we have

$$\langle \eta F_\infty(z_{t+1}) + \eta c_{t+1} - \eta F_\infty(z_t) - \eta c_t, z_{t+1} - z_t\rangle - \frac{\rho}{\eta}\|\eta F_\infty(z_{t+1}) + \eta c_{t+1} - \eta F_\infty(z_t) - \eta c_t\|^2 \geq 0 \tag{75}$$

which means

$$\begin{aligned}
&\langle \eta F_t(z_{t+1}) + \eta c_{t+1} - \eta F_{t-1}(z_t) - \eta c_t, z_{t+1} - z_t\rangle - \frac{\rho}{\eta}\|\eta F_\infty(z_{t+1}) + \eta c_{t+1} - \eta F_\infty(z_t) - \eta c_t\|^2 \\
&- \langle \eta G_t(z_{t+1}) - \eta G_{t-1}(z_t), z_{t+1} - z_t\rangle \\
&\geq 0
\end{aligned} \tag{76}$$

Since $F_\infty$ is $L$-Lipschitz, we have

$$\eta^2 L^2\left\|z_{t+1} - z_{t+\frac{1}{2}}\right\|^2 - \left\|\eta F_\infty(z_{t+1}) - \eta F_\infty(z_{t+\frac{1}{2}})\right\|^2 \geq 0 \tag{77}$$

which means

$$\eta^2 L^2 \left\| z_{t+1} - z_{t+\frac{1}{2}} \right\|^2 - \left\| \eta F_t(z_{t+1}) - \eta F_t(z_{t+\frac{1}{2}}) \right\|^2 - \left\| -\eta G_t(z_{t+1}) + \eta G_t(z_{t+\frac{1}{2}}) \right\|^2$$
$$- 2 \left\langle \eta F_t(z_{t+1}) - \eta F_t(z_{t+\frac{1}{2}}), -\eta G_t(z_{t+1}) + \eta G_t(z_{t+\frac{1}{2}}) \right\rangle \tag{78}$$
$$\geq 0$$

By multiplying the above inequality by $1 - \frac{\rho}{3\eta} > 0$, we get

$$p \left\| z_{t+1} - z_{t+\frac{1}{2}} \right\|^2 - \left\| \eta F_t(z_{t+1}) - \eta F_t(z_{t+\frac{1}{2}}) \right\|^2 + \left( \left(1 - \frac{\rho}{3\eta}\right) \eta^2 L^2 - p \right) \left\| z_{t+1} - z_{t+\frac{1}{2}} \right\|^2$$
$$+ \frac{\rho}{3\eta} \left\| \eta F_t(z_{t+1}) - \eta F_t(z_{t+\frac{1}{2}}) \right\|^2 - \left(1 - \frac{\rho}{3\eta}\right) \left\| -\eta G_t(z_{t+1}) + \eta G_t(z_{t+\frac{1}{2}}) \right\|^2$$
$$- 2 \left(1 - \frac{\rho}{3\eta}\right) \left\langle \eta F_t(z_{t+1}) - \eta F_t(z_{t+\frac{1}{2}}), -\eta G_t(z_{t+1}) + \eta G_t(z_{t+\frac{1}{2}}) \right\rangle$$
$$\geq 0$$
$$\tag{79}$$

We show the following two equations involving $z_{t+1/2}$ and $z_{t+1}$ with the update rules of the ARG algorithm.

$$z_{t+\frac{1}{2}} = 2z_t - z_{t-1} + \frac{1}{t+1}(z_0 - z_t) - \frac{1}{t}(z_0 - z_{t-1})$$
$$= z_t + (z_t - z_{t-1}) + \frac{1}{t+1}(z_0 - z_t) - \frac{1}{t}(z_0 - z_{t-1}) \tag{80}$$
$$= z_t - \eta F_{t-1}(z_{t-\frac{1}{2}}) - \eta c_t + \frac{1}{t+1}(z_0 - z_t)$$

$$z_{t+1} = z_t - \eta F_t(z_{t+\frac{1}{2}}) - \eta c_{t+1} + \frac{1}{t+1}(z_0 - z_t) \tag{81}$$

Hence, we have

$$z_{t+1} - z_{t+\frac{1}{2}} = \eta F_{t-1}(z_{t-\frac{1}{2}}) + \eta c_t - \eta F_t(z_{t+\frac{1}{2}}) - \eta c_{t+1} \tag{82}$$

Next, we simplify

$$V_t - V_{t+1} - t(t+1) \cdot \text{LHS of Inequality (76)} - \frac{t(t+1)}{4p} \cdot \text{LHS of Inequality (79)} \tag{83}$$

using the second identity in Lemma 4: replace $x_0$ with $z_0$; for $k \in [[4]]$, replace $x_k$ with $z_{t-1+\frac{k}{2}}$ and replace $y_k$ with $\eta F(z_{t-1+\frac{k}{2}})$; replace $u_2$ with $\eta c_t$; replace $u_4$ with $\eta c_{t+1}$; replace $k$ with $t$; replace $p$ with $q$. Note that $x_3 = x_2 - y_1 - u_2 + \frac{1}{k+1}(x_0 - x_2)$ and $x_4 = x_2 - y_3 - u_4 + \frac{1}{k+1}(x_0 - x_2)$ hold due to the above equivalent formations of $z_{t+1/2}$ and $z_{t+1}$.

$$V_t - V_{t+1} - t(t+1) \cdot \text{LHS of Inequality (76)} - \frac{t(t+1)}{4p} \cdot \text{LHS of Inequality (79)}$$

$$= \frac{t(t+1)}{4} \left\| \eta c_{t+1} - \eta c_t + \eta F_{t-1}(z_{t-\frac{1}{2}}) - 2\eta F_{t-1}(z_t) + \eta F_t(z_{t+\frac{1}{2}}) \right\|^2$$

$$+ \left( \frac{(1-4p)t - 4p}{4p}(t+1) \right) \left\| \eta F_t(z_{t+\frac{1}{2}}) - \eta F_t(z_{t+1}) \right\|^2 + (t+1) \left\langle \eta F_t(z_{t+\frac{1}{2}}) - \eta F_t(z_{t+1}), \eta F_t(z_{t+1}) + \eta c_{t+1} \right\rangle$$

$$+ t(t+1)\frac{\rho}{\eta} \| \eta F_\infty(z_{t+1}) + \eta c_{t+1} - \eta F_\infty(z_t) - \eta c_t \|^2$$

$$- \frac{t(t+1)}{4p} \left( \left( \left(1 - \frac{\rho}{3\eta}\right) \eta^2 L^2 - p \right) \left\| z_{t+1} - z_{t+\frac{1}{2}} \right\|^2 + \frac{\rho}{3\eta} \left\| \eta F_t(z_{t+1}) - \eta F_t(z_{t+\frac{1}{2}}) \right\|^2 \right)$$

$$+ t(t+1)\langle \eta G_t(z_{t+1}) - \eta G_{t-1}(z_t), z_{t+1} - z_t \rangle + \frac{t(t+1)}{4p} \left(1 - \frac{\rho}{3\eta}\right) \left\| -\eta G_t(z_{t+1}) + \eta G_t(z_{t+\frac{1}{2}}) \right\|^2$$

$$+ \frac{t(t+1)}{2p} \left(1 - \frac{\rho}{3\eta}\right) \left\langle \eta F_t(z_{t+1}) - \eta F_t(z_{t+\frac{1}{2}}), -\eta G_t(z_{t+1}) + \eta G_t(z_{t+\frac{1}{2}}) \right\rangle$$
$$\tag{84}$$

Since $\|a\|^2 + \langle a, b \rangle = \|a + \frac{b}{2}\|^2 - \frac{\|b\|^2}{4}$, with $p = \frac{1}{24}$, we have

$$
\left( \frac{(1-4p)t - 4p}{4p}(t+1) \right) \left\| \eta F_t(z_{t+\frac{1}{2}}) - \eta F_t(z_{t+1}) \right\|^2 + (t+1) \left\langle \eta F_t(z_{t+\frac{1}{2}}) - \eta F_t(z_{t+1}), \eta F_t(z_{t+1}) + \eta c_{t+1} \right\rangle
$$

$$
= \left\| \sqrt{\frac{(1-4p)t - 4p}{4p}(t+1)} \left( \eta F_t(z_{t+\frac{1}{2}}) - \eta F_t(z_{t+1}) \right) + \sqrt{\frac{p(t+1)}{(1-4p)t - 4p}} (\eta F_t(z_{t+1}) + \eta c_{t+1}) \right\|^2
$$

$$
- \frac{p(t+1)}{(1-4p)t - 4p} \| \eta F_t(z_{t+1}) + \eta c_{t+1} \|^2
$$

$$
\geq - \frac{p(t+1)}{(1-8p)t} \| \eta F_t(z_{t+1}) + \eta c_{t+1} \|^2 \qquad (t \geq 1)
$$

$$
\geq - \frac{2p}{1-8p} \| \eta F_t(z_{t+1}) + \eta c_{t+1} \|^2 \qquad (\frac{t+1}{t} \leq 2)
$$

$$
= - \frac{1}{8} \| \eta F_t(z_{t+1}) + \eta c_{t+1} \|^2
$$

$$\tag{85}$$

Then, we get

$$
\frac{4}{t(t+1)} \left( \frac{t(t+1)}{4} \left\| \eta c_{t+1} - \eta c_t + \eta F_{t-1}(z_{t-\frac{1}{2}}) - 2\eta F_{t-1}(z_t) + \eta F_t(z_{t+\frac{1}{2}}) \right\|^2 \right.
$$

$$
+ t(t+1) \frac{\rho}{\eta} \| \eta F_\infty(z_{t+1}) + \eta c_{t+1} - \eta F_\infty(z_t) - \eta c_t \|^2
$$

$$
\left. - \frac{t(t+1)}{4p} \left( \left( \left( 1 - \frac{\rho}{3\eta} \right) \eta^2 L^2 - p \right) \left\| z_{t+1} - z_{t+\frac{1}{2}} \right\|^2 + \frac{\rho}{3\eta} \left\| \eta F_t(z_{t+1}) - \eta F_t(z_{t+\frac{1}{2}}) \right\|^2 \right) \right)
$$

$$
= \left\| \eta c_{t+1} - \eta c_t + \eta F_{t-1}(z_{t-\frac{1}{2}}) - 2\eta F_{t-1}(z_t) + \eta F_t(z_{t+\frac{1}{2}}) \right\|^2 + \left( 1 - \left( 24 - \frac{8\rho}{\eta} \right) \eta^2 L^2 \right) \left\| z_{t+1} - z_{t+\frac{1}{2}} \right\|^2
$$

$$
+ \frac{4\rho}{\eta} \| \eta F_\infty(z_{t+1}) + \eta c_{t+1} - \eta F_\infty(z_t) - \eta c_t \|^2 - \frac{8\rho}{\eta} \left\| \eta F_t(z_{t+1}) - \eta F_t(z_{t+\frac{1}{2}}) \right\|^2
$$

$$\tag{86}$$

When there exists

$$
\begin{aligned}
B_1 &= \eta c_{t+1} - \eta c_t + \eta F_{t-1}(z_{t-\frac{1}{2}}) - 2\eta F_{t-1}(z_t) + \eta F_t(z_{t+\frac{1}{2}}) \\
B_2 &= z_{t+1} - z_{t+\frac{1}{2}} = \eta F_{t-1}(z_{t-\frac{1}{2}}) + \eta c_t - \eta F_t(z_{t+\frac{1}{2}}) - \eta c_{t+1} \\
B_3 &= \eta F_t(z_{t+1}) + \eta c_{t+1} - \eta F_{t-1}(z_t) - \eta c_t \\
B_4 &= \eta F_t(z_{t+1}) - \eta F_t(z_{t+\frac{1}{2}})
\end{aligned}
\tag{87}
$$

we have

$$
B_1 - B_2 = 2\eta c_{t+1} - 2\eta c_t - 2\eta F_{t-1}(z_t) + 2\eta F_t(z_{t+\frac{1}{2}}) = 2(B_3 - B_4) \tag{88}
$$

Note that $\rho$ is non-positive and we have

$$
\frac{4}{t(t+1)} \left( \frac{t(t+1)}{4} \left\| \eta c_{t+1} - \eta c_t + \eta F_{t-1}(z_{t-\frac{1}{2}}) - 2\eta F_{t-1}(z_t) + \eta F_t(z_{t+\frac{1}{2}}) \right\|^2 \right.
$$

$$
+ t(t+1) \frac{\rho}{\eta} \| \eta F_\infty(z_{t+1}) + \eta c_{t+1} - \eta F_\infty(z_t) - \eta c_t \|^2
$$

$$
\left. - \frac{t(t+1)}{4p} \left( \left( \left( 1 - \frac{\rho}{3\eta} \right) \eta^2 L^2 - p \right) \left\| z_{t+1} - z_{t+\frac{1}{2}} \right\|^2 + \frac{\rho}{3\eta} \left\| \eta F_t(z_{t+1}) - \eta F_t(z_{t+\frac{1}{2}}) \right\|^2 \right) \right)
$$

$$= \|B_1\|^2 + \left(1 - \left(24 - \frac{8\rho}{\eta}\right)\eta^2 L^2\right)\|B_2\|^2 + \frac{\rho}{\eta}\|2B_3 - 2G_t(z_{t+1}) + 2G_{t-1}(z_t)\|^2 - \frac{2\rho}{\eta}\|2B_4\|^2$$

$$\geq \left(\frac{1}{2} - \left(12 - \frac{4\rho}{\eta}\right)\eta^2 L^2\right)\|B_1 - B_2\|^2 + \frac{\rho}{\eta}\|2B_3 - 2G_t(z_{t+1}) + 2G_{t-1}(z_t)\|^2 - \frac{2\rho}{\eta}\|2B_4\|^2$$

$$\left(\|a\|^2 + \|b\|^2 \geq \frac{1}{2}\|a - b\|^2 \text{ and } \left(24 - \frac{8\rho}{\eta}\right)\eta^2 L^2 \geq 0\right)$$

$$\geq \left(\frac{1}{2} - \left(12 - \frac{4\rho}{\eta}\right)\eta^2 L^2\right)\|B_1 - B_2\|^2 + \frac{2\rho}{\eta}\|2B_3 - 2B_4 - 2G_t(z_{t+1}) + 2G_{t-1}(z_t)\|^2$$

$$\left(-\|a\|^2 + 2\|b\|^2 \geq -2\|a - b\|^2 \text{ and } -\frac{\rho}{\eta} \geq 0\right)$$

$$= \left(\frac{1}{2} - \left(12 - \frac{4\rho}{\eta}\right)\eta^2 L^2 + \frac{2\rho}{\eta}\right)\|B_1 - B_2\|^2 + \frac{4\rho}{\eta}\langle 2(B_3 - B_4), -2G_t(z_{t+1}) + 2G_{t-1}(z_t)\rangle$$

$$+ \frac{2\rho}{\eta}\| - 2G_t(z_{t+1}) + 2G_{t-1}(z_t)\|^2$$

$$\geq 0$$

($\eta$ is chosen as shown in Lemma 9)

$$(89)$$

Hence, we have

$$V_t - V_{t+1}$$
$$\geq -\frac{1}{8}\|\eta F_t(z_{t+1}) + \eta c_{t+1}\|^2 + t(t+1)\langle \eta G_t(z_{t+1}) - \eta G_{t-1}(z_t), z_{t+1} - z_t\rangle$$
$$+ \frac{t(t+1)}{4p}\left(1 - \frac{\rho}{3\eta}\right)\left\|-\eta G_t(z_{t+1}) + \eta G_t(z_{t+\frac{1}{2}})\right\|^2$$
$$+ \frac{t(t+1)}{2p}\left(1 - \frac{\rho}{3\eta}\right)\left\langle \eta F_t(z_{t+1}) - \eta F_t(z_{t+\frac{1}{2}}), -\eta G_t(z_{t+1}) + \eta G_t(z_{t+\frac{1}{2}})\right\rangle$$

$$(90)$$

$\square$

### F.3 FINAL PROOF

We first show that the potential function $V_t = O(t^2 \cdot r^{tan}(z_t)^2)$.

**Lemma 11.** *With the settings of Theorem 4, $\forall t \geq 1$, we have*

$$\frac{t(t + \frac{1}{2})}{4}\|\eta F_{t-1}(z_t) + \eta c_t\|^2$$
$$\leq V_t + \|z^* - z_0\|^2 + \frac{t}{4}\|\eta G_{t-1}(z_t)\|^2 + \frac{t}{2}\langle \eta F_{t-1}(z_t) + \eta c_t, -\eta G_{t-1}(z_t)\rangle - t\langle \eta G_{t-1}(z_t), z_t - z^*\rangle$$

$$(91)$$

*Proof.* Since $F_\infty(z^*) + N_{\mathcal{Z}}(z^*) = 0$, by the settings in Theorem 4 and Lemma 9, we have

$$\langle \eta F_\infty(z_t) + \eta c_t, z_t - z^*\rangle \geq \frac{\rho}{\eta}\|\eta F_\infty(z_t) + \eta c_t\|^2 \geq -\frac{1}{4}\|\eta F_\infty(z_t) + \eta c_t\|^2 \tag{92}$$

With definition of $V_t$ in Equation (70), $\forall t \geq 1$, we obtain

$$V_t$$
$$= \frac{t(t+1)}{2}\|\eta F_{t-1}(z_t) + \eta c_t\|^2 + t\langle \eta F_{t-1}(z_t) + \eta c_t, z_t - z_0\rangle + \frac{t(t+1)}{2}\left\|\eta F_{t-1}(z_t) - \eta F_{t-1}(z_{t-\frac{1}{2}})\right\|^2$$
$$\geq \frac{t(t+1)}{2}\|\eta F_{t-1}(z_t) + \eta c_t\|^2 + t\langle \eta F_{t-1}(z_t) + \eta c_t, z_t - z^*\rangle + t\langle \eta F_{t-1}(z_t) + \eta c_t, z^* - z_0\rangle$$

$$\geq \frac{t(t+1)}{2}\|\eta F_{t-1}(z_t) + \eta c_t\|^2 - \frac{t}{4}\|\eta F_\infty(z_t) + \eta c_t\|^2 + t\langle \eta G_{t-1}(z_t), z_t - z^*\rangle + t\langle \eta F_{t-1}(z_t) + \eta c_t, z^* - z_0\rangle$$

$$\geq \frac{t(t+1)}{2}\|\eta F_{t-1}(z_t) + \eta c_t\|^2 - \frac{t}{4}\|\eta F_{t-1}(z_t) + \eta c_t\|^2 - \frac{t}{4}\|-\eta G_{t-1}(z_t)\|^2 - \frac{t}{2}\langle \eta F_{t-1}(z_t) + \eta c_t, -\eta G_{t-1}(z_t)\rangle$$

$$- \frac{t(t+\frac{1}{2})}{4}\|\eta F_{t-1}(z_t) + \eta c_t\|^2 - \frac{t}{t+\frac{1}{2}}\|z^* - z_0\|^2 + t\langle \eta G_{t-1}(z_t), z_t - z^*\rangle$$

$$\geq \frac{t(t+\frac{1}{2})}{4}\|\eta F_{t-1}(z_t) + \eta c_t\|^2 - \|z^* - z_0\|^2 - \frac{t}{4}\|\eta G_{t-1}(z_t)\|^2 - \frac{t}{2}\langle \eta F_{t-1}(z_t) + \eta c_t, -\eta G_{t-1}(z_t)\rangle$$

$$+ t\langle \eta G_{t-1}(z_t), z_t - z^*\rangle \tag{93}$$

where in the second last inequality we apply $\langle a, b\rangle \geq -\frac{\alpha}{4}\|a\|^2 - \frac{1}{\alpha}\|b\|^2$ with $a = \sqrt{t}(\eta F_t(z_t) + \eta c_t), b = \sqrt{t}(z^* - z_0)$, and $\alpha = t + \frac{1}{2}$. $\qquad\square$

*Proof of Theorem 4.* There exists $H > 0$ satisfying

$$\|\eta F_0(z_1) + \eta c_1\|^2 \leq H^2 \tag{94}$$

So the theorem holds for $T = 1$. $\forall T \geq 2$, with Lemma 11, we have

$$\frac{T(T+\frac{1}{2})}{4}\|\eta F(z_T) + \eta c_T\|^2$$

$$\leq V_T + \|z_0 - z^*\|^2 - \left(-\frac{t}{4}\|\eta G_{t-1}(z_t)\|^2 - \frac{t}{2}\langle \eta F_{t-1}(z_t) + \eta c_t, -\eta G_{t-1}(z_t)\rangle + t\langle \eta G_{t-1}(z_t), z_t - z^*\rangle\right)$$

$$\leq V_T + \|z_0 - z^*\|^2 + \frac{t}{4}\|\eta G_{t-1}(z_t)\|^2 + \frac{t}{2}\langle \eta F_{t-1}(z_t) + \eta c_t, -\eta G_{t-1}(z_t)\rangle - t\langle \eta G_{t-1}(z_t), z_t - z^*\rangle$$

$$\leq V_1 + \|z_0 - z^*\|^2 + \frac{t}{4}\|\eta G_{t-1}(z_t)\|^2 + \frac{t}{2}\langle \eta F_{t-1}(z_t) + \eta c_t, -\eta G_{t-1}(z_t)\rangle - t\langle \eta G_{t-1}(z_t), z_t - z^*\rangle$$

$$+ \frac{1}{8}\sum_{t=2}^{T}\|\eta F(z_t) + \eta c_t\|^2 - \sum_{t=1}^{T-1} t(t+1)\langle \eta G_t(z_{t+1}) - \eta G_{t-1}(z_t), z_{t+1} - z_t\rangle$$

$$- \sum_{t=1}^{T-1}\frac{t(t+1)}{4p}\left(1 - \frac{\rho}{3\eta}\right)\left\|-\eta G_t(z_{t+1}) + \eta G_t(z_{t+\frac{1}{2}})\right\|^2$$

$$- \sum_{t=1}^{T-1}\frac{t(t+1)}{2p}\left(1 - \frac{\rho}{3\eta}\right)\left\langle \eta F_t(z_{t+1}) - \eta F_t(z_{t+\frac{1}{2}}), -\eta G_t(z_{t+1}) + \eta G_t(z_{t+\frac{1}{2}})\right\rangle \tag{95}$$

By subtracting $\frac{1}{8}\|\eta F(z_T) + \eta c_T\|^2$ from both sides of the above inequality, we get

$$\frac{T^2}{4}\|\eta F_{T-1}(z_T) + \eta c_T\|^2$$

$$\leq \frac{1}{8}\sum_{t=2}^{T-1}\|\eta F_{t-1}(z_t) + \eta c_t\|^2 + \|z_0 - z^*\|^2 + \frac{t}{4}\|\eta G_{t-1}(z_t)\|^2 + \frac{t}{2}\langle \eta F_{t-1}(z_t) + \eta c_t, -\eta G_{t-1}(z_t)\rangle$$

$$- t\langle \eta G_{t-1}(z_t), z_t - z^*\rangle - \sum_{t=1}^{T-1} t(t+1)\langle \eta G_t(z_{t+1}) - \eta G_{t-1}(z_t), z_{t+1} - z_t\rangle$$

$$- \sum_{t=1}^{T-1}\frac{t(t+1)}{4p}\left(1 - \frac{\rho}{3\eta}\right)\left\|-\eta G_t(z_{t+1}) + \eta G_t(z_{t+\frac{1}{2}})\right\|^2$$

$$- \sum_{t=1}^{T-1}\frac{t(t+1)}{2p}\left(1 - \frac{\rho}{3\eta}\right)\left\langle \eta F_t(z_{t+1}) - \eta F_t(z_{t+\frac{1}{2}}), -\eta G_t(z_{t+1}) + \eta G_t(z_{t+\frac{1}{2}})\right\rangle \tag{96}$$

Since Assumption 1 holds, we have $\max\|G_t\| = O(\frac{1}{t})$.

We show that there exists $D_1, D_2 > 0$ satisfying

$$S_T = \|\eta F_{T-1}(z_T) + \eta c_T\| \leq \frac{D_1}{T} + D_2 \sum_{t=2}^{T} \max \|G_{t-1}\| \tag{97}$$

as follows. There exists $D_1, D_2 \geq 0$ satisfying

$$S_2 \leq \frac{D_1}{2} + D_2 \max \|G_1\| \tag{98}$$

$$S_T \leq \frac{1}{2T^2} \sum_{t=2}^{T-1} S_{t-1} + D_2 \sum_{t=2}^{T-1} \max \|G_{t-1}\| \tag{99}$$

If $D_1 > 0$ and $D_1 \geq D_2 \sum_{t=2}^{\infty} \max \|G_{t-1}\|$, with the assumption that Inequality (97) holds true when $T$ is replaced by $2, 3, \cdots, T-1$, we have

$$
\begin{aligned}
S_T &\leq \frac{D_1}{2T^2} \sum_{t=2}^{T-1} \frac{1}{t-1} + \frac{D_2}{2T^2} \sum_{t=2}^{T-1} \sum_{s=2}^{t} \max \|G_{s-1}\| + D_2 \sum_{t=2}^{T-1} \max \|G_{t-1}\| \\
&\leq \frac{D_1(T-2)}{2T^2} + \frac{D_2(T-2)}{2T^2} \sum_{s=2}^{T-1} \max \|G_{s-1}\| + D_2 \sum_{t=2}^{T-1} \max \|G_{t-1}\| \\
&\leq \frac{D_1(T-2)}{T^2} + D_2 \sum_{t=2}^{T-1} \max \|G_{t-1}\| \\
&\leq \frac{D_1}{T} + D_2 \sum_{t=2}^{T-1} \max \|G_{t-1}\|
\end{aligned}
\tag{100}
$$

Since there exists $D_1 \geq D_2 \sum_{t=2}^{\infty} \max \|G_{t-1}\|$ satisfying (98), Inequality (97) holds true.

Fianlly, we show that there exists $D_3, D_4 > 0$ satisfying

$$S_T = \|\eta F_{T-1}(z_T) + \eta c_T\| \leq \frac{D_3}{T} \tag{101}$$

as follows. With the assumption that Inequality (101) holds true when $T$ is replaced by $2, 3, \cdots, T-1$, we have

$$
\begin{aligned}
S_T &\leq \frac{D_1}{2T^2} \sum_{t=2}^{T-1} \frac{1}{t-1} + \frac{D_2}{2T^2} \sum_{t=2}^{T-1} \sum_{s=2}^{t} \max \|G_{s-1}\| + D_2 \sum_{t=2}^{T-1} \max \|G_{t-1}\| \\
&\leq \frac{D_1(T-2)}{2T^2} + \frac{D_2(T-2)}{2T^2} \sum_{s=2}^{T-1} \max \|G_{s-1}\| + D_2 \sum_{t=2}^{T-1} \max \|G_{t-1}\| \\
&\leq \frac{D_1(T-2)}{T^2} + D_2 \sum_{t=2}^{T-1} \max \|G_{t-1}\| \\
&\leq \frac{D_1}{T} + D_2 \sum_{t=2}^{T-1} \max \|G_{t-1}\|
\end{aligned}
\tag{102}
$$

If $\sum_{t=0}^{\infty} t^2 \|G_t\| < \infty$, there exists $D_4 > 0$ satisfying

$$S_T \leq \frac{1}{2T^2} \left( \sum_{t=2}^{T-1} S_{t-1} + D_4 \right) \tag{103}$$

If Inequality (101) holds when $T$ is replaced by $2, 3, \cdots, T-1$, then

$$S_T \leq \frac{D_3}{2T^2} \left( \sum_{t=2}^{T-1} \frac{1}{t-1} + D_4 \right) \leq \frac{D_3}{2T} + \frac{D_3}{2T^2 D_4} = \frac{D_3}{T} \left( \frac{1}{2} + \frac{1}{2 D_4 T} \right) \leq \frac{D_3}{T} \tag{104}$$

When $D_3 = 2S_2$, Inequality (101) holds with $T$ replaced by 2. Hence, Inequality (101) holds with $D_3 = 2S_2$.

This completes the proof. $\qquad\square$

## G   MISSING PROOFS OF THEOREM 5

We apply the modified OG algorithm to the two-player bilinear game (1) shown in Theorem 1.

For the general two-player bilinear game

$$
\begin{aligned}
z^{(1)} &= x \in \mathbb{R}^m, \\
z^{(2)} &= y \in \mathbb{R}^n, \\
f^{(1)}(z) &= x^T A_t y, \\
f^{(2)}(z) &= -x^T A_t y, \\
\mathcal{Z} &= \mathbb{R}^m \times \mathbb{R}^n
\end{aligned}
\tag{105}
$$

with the RG algorithm, there exists

$$
\begin{bmatrix} x_t \\ y_t \\ x_{t+1} \\ y_{t+1} \end{bmatrix}
=
\begin{bmatrix}
0 & 0 & I & 0 \\
0 & 0 & 0 & I \\
0 & \eta A_t & I & -2\eta A_t \\
-\eta A_t^T & 0 & 2\eta A_t^T & I
\end{bmatrix}
\begin{bmatrix} x_{t-1} \\ y_{t-1} \\ x_t \\ y_t \end{bmatrix}
\tag{106}
$$

With the modified OG algorithm, there exists

$$
\begin{bmatrix} x_{t+\frac{1}{2}} \\ y_{t+\frac{1}{2}} \\ x_{t+\frac{3}{2}} \\ y_{t+\frac{3}{2}} \end{bmatrix}
=
\begin{bmatrix}
0 & 0 & I & 0 \\
0 & 0 & 0 & I \\
0 & \eta A_{t+1} & I & -2\eta A_{t+1} \\
-\eta A_{t+1}^T & 0 & 2\eta A_{t+1}^T & I
\end{bmatrix}
\begin{bmatrix} x_{t-\frac{1}{2}} \\ y_{t-\frac{1}{2}} \\ x_{t+\frac{1}{2}} \\ y_{t+\frac{1}{2}} \end{bmatrix}
\tag{107}
$$

The equivalence of the RG and the variant of the OG algorithm algorithms is proven.

In the settings provided in Theorem 5, with the modified OG algorithm, as the relation shown in (107), we have

$$
\begin{bmatrix} x_{t+\frac{1}{2}} \\ y_{t+\frac{1}{2}} \\ x_{t+\frac{3}{2}} \\ y_{t+\frac{3}{2}} \end{bmatrix}
=
\begin{bmatrix}
0 & 0 & I & 0 \\
0 & 0 & 0 & I \\
0 & \eta A_{t+1} & I & -2\eta A_{t+1} \\
-\eta A_{t+1}^T & 0 & 2\eta A_{t+1}^T & I
\end{bmatrix}
\begin{bmatrix} x_{t-\frac{1}{2}} \\ y_{t-\frac{1}{2}} \\ x_{t+\frac{1}{2}} \\ y_{t+\frac{1}{2}} \end{bmatrix}
\tag{108}
$$

Hence, with the notation in the proof of Theorem 1, we have

$$
\begin{bmatrix} x_{t-\frac{1}{2}} \\ y_{t-\frac{1}{2}} \\ x_{t+\frac{1}{2}} \\ y_{t+\frac{1}{2}} \end{bmatrix}
= P_t
\begin{bmatrix} x_{t-\frac{3}{2}} \\ y_{t-\frac{3}{2}} \\ x_{t-\frac{1}{2}} \\ y_{t-\frac{1}{2}} \end{bmatrix}
\tag{109}
$$

Under the modified OG algorithm, there exists $z_0$, $z_{-1/2}$ satisfying

$$
\begin{bmatrix} z_{-\frac{1}{2}} \\ z_{\frac{1}{2}} \end{bmatrix}
=
\begin{bmatrix} x_{-\frac{1}{2}} \\ y_{-\frac{1}{2}} \\ x_{\frac{1}{2}} \\ y_{\frac{1}{2}} \end{bmatrix}
= c
\tag{110}
$$

where $c$ is defined in Theorem 1. Without loss of generality, assume $t$ is odd. Then, we have

$$
\begin{bmatrix} z_{t+\frac{1}{2}} \\ z_{t+\frac{3}{2}} \end{bmatrix}
= P_{t+1} P_t
\begin{bmatrix} z_{t-\frac{3}{2}} \\ z_{t-\frac{1}{2}} \end{bmatrix}
\tag{111}
$$

Since $P_{t+1} P_t$ has an eigenvalue $\lambda_{1,2} = \frac{8\eta^2 + 1 + \sqrt{64\eta^4 + 8\eta^2 + 1}}{2} > 1$, there exists a corresponding eigenvector

$$
\begin{bmatrix}
\frac{16\eta^2 - 2\sqrt{64\eta^4 + 8\eta^2 + 1} + 4}{2(8\eta^2 + 1)} \\
\frac{\sqrt{64\eta^4 + 8\eta^2 + 1} - 1}{4\eta(8\eta^2 + 1)} \\
-\frac{\sqrt{64\eta^4 + 8\eta^2 + 1} - 1}{4\eta(8\eta^2 + 1)} \\
1 \\
0 \\
0
\end{bmatrix}
\tag{112}
$$

If $c$ is equal to this eigenvector, we have $\|z_{1/2}\| = 1$ and $\|z_{-1/2}\| = \frac{\sqrt{8\eta^2 - \sqrt{64\eta^4 + 8\eta^2 + 1} + 1}}{2\eta}$. Since $8\eta^2 - \sqrt{64\eta^4 + 8\eta^2 + 1} + 1 = 8\eta^2 + \frac{1}{2} - \sqrt{64\eta^4 + 8\eta^2 + 1} + \frac{1}{2} = \sqrt{64\eta^4 + 8\eta^2 + \frac{1}{4}} - \sqrt{64\eta^4 + 8\eta^2 + 1} + \frac{1}{2} < \frac{1}{2}, \forall \eta > 0, \eta\|z_{-1/2}\| < \|z_{1/2}\|$. With such $z_0$ and $z_{-1/2}$, $\|z_{t-1/2}\|$ and $\|z_{t+1/2}\|$ diverges exponentially. Since

$$\begin{bmatrix} x_t \\ y_t \end{bmatrix} = \begin{bmatrix} x_{t+\frac{1}{2}} \\ y_{t+\frac{1}{2}} \end{bmatrix} + \eta \begin{bmatrix} A_t y_{t-\frac{1}{2}} \\ -A_t^T x_{t-\frac{1}{2}} \end{bmatrix} \tag{113}$$

we have

$$\|z_t\| = \|z_{t+\frac{1}{2}} + F_t(z_{t-\frac{1}{2}})\| \geq \|\|z_{t+\frac{1}{2}}\| - \|F_t(z_{t-\frac{1}{2}})\|\| = \|\|z_{t+\frac{1}{2}}\| - \eta\|z_{t-\frac{1}{2}}\|\| \tag{114}$$

Since $\|z_{1/2}\| - \eta\|z_{-1/2}\| \neq 0$ in the $c$ equal to (112), for $t = 2k - 1$, there exists

$$\|r^{tan}(z_{2k-1})\| = \|F_{2k-1}(z_{2k-1})\| = \|z_{2k-1}\| \geq \|\|z_{2k+\frac{1}{2}}\| - \eta\|z_{2k-\frac{1}{2}}\|\| = \lambda_{1,2}^k\|\|z_{\frac{1}{2}}\| - \eta\|z_{-\frac{1}{2}}\|\| > 0 \tag{115}$$

which means $\|r^{tan}(z_{2k-1})\|$ also diverges at an exponential rate. This completes the proof.

## H  MISSING PROOFS OF THEOREM 6

We only need to prove the extended result. With the update rules of the OG algorithm, we have the following identity (Hsieh et al. (2019)): $\forall p \in \mathcal{Z}$,

$$\|z_{t+1} - p\|^2 = \|z_t - p\|^2 + \left\|z_{t+1} - z_{t+\frac{1}{2}}\right\|^2 - \left\|z_{t+\frac{1}{2}} - z_t\right\|^2 + 2\left\langle z_t - \eta F_t(z_{t-\frac{1}{2}}) - z_{t+\frac{1}{2}} + \eta F_t(z_{t+\frac{1}{2}}), p - z_{t+\frac{1}{2}}\right\rangle \tag{116}$$

Since $z_{t+1/2} = \Pi_{\mathcal{Z}}(z_t - \eta F_t(z_{t-1/2}))$, we have

$$\frac{z_t - \eta F_t(z_{t-\frac{1}{2}}) - z_{t+\frac{1}{2}}}{\eta} \in N_{\mathcal{Z}}(z_{t+\frac{1}{2}}) \tag{117}$$

Then

$$\frac{z_t - z_{t+1}}{\eta} = \frac{z_t - \eta F_t(z_{t-\frac{1}{2}}) - z_{t+\frac{1}{2}}}{\eta} + F_t(z_{t+\frac{1}{2}}) \in F_t(z_{t+\frac{1}{2}}) + N_{\mathcal{Z}}(z_{t+\frac{1}{2}}) \tag{118}$$

which means

$$r_{F,\mathcal{Z}}^{tan}(z_{t+\frac{1}{2}}) \leq \frac{\|z_{t+1} - z_t\|}{\eta} \tag{119}$$

For $p = z^*$, we have

$$2\left\langle z_t - \eta F_t(z_{t-\frac{1}{2}}) - z_{t+\frac{1}{2}} + \eta F_t(z_{t+\frac{1}{2}}), z^* - z_{t+\frac{1}{2}}\right\rangle = 2\eta\left\langle \frac{z_t - z_{t+1}}{\eta}, z^* - z_{t+\frac{1}{2}}\right\rangle \leq -\frac{2\rho}{\eta}\|z_t - z_{t+1}\|^2 \tag{120}$$

Define $c = \frac{1 - 2L_t^2\eta^2 - 2L_{t-1}^2\eta^2}{2(2L_t^2\eta^2 - 2L_{t-1}^2\eta^2 + 1)} > 0$. We have identity

$$(1 - 2c)\eta^2 L_{t-1}^2 = \frac{1}{2} - c - (1 + 2c)\eta^2 L_t^2 \tag{121}$$

Combining (116) and (120), since $\|a + b\|^2 \leq 2\|a\|^2 + 2\|b\|^2$, we have

$$\|z_{t+1} - z^*\|^2$$
$$\leq \|z_t - z^*\|^2 + \left\|z_{t+1} - z_{t+\frac{1}{2}}\right\|^2 - \left\|z_{t+\frac{1}{2}} - z_t\right\|^2 + c\|z_t - z_{t+1}\|^2 - \left(c + \frac{2\rho}{\eta}\right)\|z_t - z_{t+1}\|^2$$
$$\leq \|z_t - z^*\|^2 + (1 + 2c)\left\|z_{t+1} - z_{t+\frac{1}{2}}\right\|^2 - (1 - 2c)\left\|z_{t+\frac{1}{2}} - z_t\right\|^2 - \left(c + \frac{2\rho}{\eta}\right)\|z_t - z_{t+1}\|^2 \tag{122}$$

Using the update rules of OG and $L_t$-Lipschitzness of $F_t$, we have that for any $t \geq 0$,

$$\left\| z_{t+1} - z_{t+\frac{1}{2}} \right\|^2 = \left\| \eta F_t(z_{t-\frac{1}{2}}) - \eta F_t(z_{t+\frac{1}{2}}) \right\|^2 \leq \eta^2 L_t^2 \left\| z_{t+\frac{1}{2}} - z_{t-\frac{1}{2}} \right\|^2 \tag{123}$$

Moreover, using $\|a + b\|^2 \leq 2\|a\|^2 + 2\|b\|^2$ and (123), we have that for any $t \geq 1$,

$$\left\| z_{t+\frac{1}{2}} - z_{t-\frac{1}{2}} \right\|^2 \leq 2 \left\| z_{t+\frac{1}{2}} - z_t \right\|^2 + 2 \left\| z_t - z_{t-\frac{1}{2}} \right\|^2 \leq 2 \left\| z_{t+\frac{1}{2}} - z_t \right\|^2 + 2\eta^2 L_{t-1}^2 \left\| z_{t-\frac{1}{2}} - z_{t-\frac{3}{2}} \right\|^2 \tag{124}$$

which means

$$\left\| z_{t+\frac{1}{2}} - z_t \right\|^2 \geq \frac{1}{2} \left\| z_{t+\frac{1}{2}} - z_{t-\frac{1}{2}} \right\|^2 - \eta^2 L_{t-1}^2 \left\| z_{t-\frac{1}{2}} - z_{t-\frac{3}{2}} \right\|^2 \tag{125}$$

Hence,

$$\|z_{t+1} - z^*\|^2$$

$$\leq \|z_t - z^*\|^2 + (1 + 2c)\|z_{t+1} - z_{t+\frac{1}{2}}\|^2 - (1 - 2c) \left\| z_{t+\frac{1}{2}} - z_t \right\|^2 - \left( c + \frac{2\rho}{\eta} \right) \|z_t - z_{t+1}\|^2$$

$$\leq \|z_t - z^*\|^2 + (1 - 2c)\eta^2 L_{t-1}^2 \left\| z_{t-\frac{1}{2}} - z_{t-\frac{3}{2}} \right\|^2 - \left( \frac{1}{2} - c - (1 + 2c)\eta^2 L_t^2 \right) \left\| z_{t+\frac{1}{2}} - z_{t-\frac{1}{2}} \right\|^2$$

$$\quad - \left( c + \frac{2\rho}{\eta} \right) \|z_t - z_{t+1}\|^2$$

$$= \|z_t - z^*\|^2 + \frac{4L_t^2 L_{t-1}^2 \eta^4}{2L_t^2 \eta^2 - 2L_{t-1}^2 \eta^2 + 1} \left( \left\| z_{t-\frac{1}{2}} - z_{t-\frac{3}{2}} \right\|^2 - \left\| z_{t+\frac{1}{2}} - z_{t-\frac{1}{2}} \right\|^2 \right) - \left( c + \frac{2\rho}{\eta} \right) \|z_t - z_{t+1}\|^2 \tag{126}$$

By summing the above inequality for $t = 1, 2, \cdots, T$, with $c = \frac{1 - 2L_t^2 \eta^2 - 2L_{t-1}^2 \eta^2}{2(2L_t^2 \eta^2 - 2L_{t-1}^2 \eta^2 + 1)}$, considering $L_s$-Lipschitz means $L_g$-Lipschitz if $L_s < L_g$, $L_t$ with $t = 1, 2, \cdots, T$ can be replaced by $L_{\max}$. Hence, we get

$$\left( \frac{1}{2} - 2\eta^2 L_{\max}^2 + \frac{2\rho}{\eta} \right) \sum_{t=1}^{T} \|z_t - z_{t+1}\|^2 \leq \|z_1 - z^*\|^2 + \frac{1}{4} \left\| z_{\frac{1}{2}} - z_{-\frac{1}{2}} \right\|^2 \tag{127}$$

which means

$$\min_{t \in [[T]]} \left( \frac{\|z_{t+1} - z_t\|}{\eta} \right)^2 = O\left( \frac{1}{T} \right) \tag{128}$$

Hence, we have

$$\min_{t \in [[T]]} r_{F,\mathcal{Z}}^{tan}(z_{t+\frac{1}{2}}) \leq \min_{t \in [[T]]} \frac{\|z_{t+1} - z_t\|}{\eta} = O\left( \frac{1}{\sqrt{T}} \right) \tag{129}$$

This completes the proof.

# I    NUMERICAL EXAMPLES

Several visual representations are provided to illustrate and verify the theoretical results, i.e., Theorem 1, Theorem 2, Theorem 3, Theorem 4, Theorem 5 and Theorem 6. These examples are based on examples of bilinear games shown in Feng et al. (2023). Running all the programs for the following experiments costs no more than dozens of minutes in total with the corresponding code and the computer resources of an ordinary laptop.

## I.1    EXPERIMENTS ON THEOREM 1

We verify Theorem 1 by the example

$$z^{(1)} = x \in \mathbb{R}, z^{(2)} = y \in \mathbb{R}^2$$

$$f^{(1)} = x^T A_t y, f^{(2)} = -x^T A_t y \tag{130}$$

$$\mathcal{Z} = \mathbb{R} \times \mathbb{R}^2$$

where

$$A_t = \begin{cases} \begin{bmatrix} 1 & -1 \end{bmatrix}, t \text{ is odd} \\ \begin{bmatrix} -1 & 1 \end{bmatrix}, t \text{ is even} \end{cases} \tag{131}$$

The step size $\eta$ is chosen to be 0.005. The initial points are chosen to be $x_0 = 1, x_{-1} = 4$ and $y_0 = [2, 3], y_{-1} = [5, 6]$. The experimental results are presented in Figure 1, where we can see the RG algorithm makes $r^{tan}(z_t)$ diverge. This result supports the exponential divergence result in Theorem 1.

### I.2 EXPERIMENTS ON THEOREM 2

We verify Theorem 2 by the example

$z^{(1)} = x \in [-100, 100]^2, z^{(2)} = y \in [-100, 100]^2$

$f^{(1)}(z) = (x^T - [2 \quad 3]) \begin{bmatrix} 2 & 3 \\ 4 & 6 \end{bmatrix} \left( y - \begin{bmatrix} 5 \\ 3 \end{bmatrix} \right) + 100t^i(-25\sin(z_1) + 30\sin(z_2) + 60\sin(z_3) + 50\sin(z_4))$

$f^{(2)}(z) = -(x^T - [2 \quad 3]) \begin{bmatrix} 2 & 3 \\ 4 & 6 \end{bmatrix} \left( y - \begin{bmatrix} 5 \\ 3 \end{bmatrix} \right) + 100t^i(-25\sin(z_1) + 30\sin(z_2) + 60\sin(z_3) + 50\sin(z_4))$

$\mathcal{Z} = [-100, 100]^4$
$$\tag{132}$$

where $i = -1.1, -2, -9$ in three cases. The step size is chosen to be 0.005. The initial points are chosen to be $x_0 = [15, 40], x_{-1} = [44, 35]$ and $y_0 = [3, 51], y_{-1} = [3, 21]$. The experimental results are presented in Figure 2, all of the three dynamics make $r^{tan}(z_t)$ converge to 0, thus support the convergence result in Theorem 2.

### I.3 EXPERIMENTS ON THEOREM 3

We verify Theorem 3 by the example

$$z^{(1)} = x \in \mathbb{R}, z^{(2)} = y \in \mathbb{R}^2$$
$$f^{(1)}(z) = x^T A_t y, f^{(2)}(z) = -x^T A_t y \tag{133}$$
$$\mathcal{Z} = \mathbb{R} \times \mathbb{R}^2$$

where

$$A_t = \begin{cases} \begin{bmatrix} 1 & -1 \end{bmatrix}, t \text{ is odd} \\ \begin{bmatrix} -1 & 1 \end{bmatrix}, t \text{ is even} \end{cases} \tag{134}$$

The step size $\eta$ is chosen to be 0.005. The initial points are chosen to be $x_0 = 1, x_{-\frac{1}{2}} = 4$ and $y_0 = [2, 3], y_{-\frac{1}{2}} = [5, 6]$. The experimental results are presented in Figure 3, where we can see the ARG algorithm makes $r^{tan}(z_t)$ diverge. This result supports the exponentially divergence result in Theorem 3.

### I.4 EXPERIMENTS ON THEOREM 4

We verify Theorem 4 by the example

$z^{(1)} = x \in [-100, 100]^2, z^{(2)} = y \in [-100, 100]^2$

$f^{(1)}(z) = (x^T - [2 \quad 3]) \begin{bmatrix} 2 & 3 \\ 4 & 6 \end{bmatrix} \left( y - \begin{bmatrix} 5 \\ 3 \end{bmatrix} \right) + 100t^i(-25\sin(z_1) + 30\sin(z_2) + 60\sin(z_3) + 50\sin(z_4))$

$f^{(2)}(z) = -(x^T - [2 \quad 3]) \begin{bmatrix} 2 & 3 \\ 4 & 6 \end{bmatrix} \left( y - \begin{bmatrix} 5 \\ 3 \end{bmatrix} \right) + 100t^i(-25\sin(z_1) + 30\sin(z_2) + 60\sin(z_3) + 50\sin(z_4))$

$\mathcal{Z} = [-100, 100]^4$
$$\tag{135}$$

where $i = -1.1, -2, -9$ in three cases. The step size is chosen to be 0.005. The initial points are chosen to be $x_0 = [15, 40], x_{-1} = [44, 35]$ and $y_0 = [3, 51], y_{-1} = [3, 21]$. The experimental results are presented in Figure 4, all of the three dynamics make $r^{tan}(z_t)$ converge to 0, thus support the convergence result in Theorem 4.

## I.5 EXPERIMENTS ON THEOREM 5

We verify Theorem 5 by the example

$$z^{(1)} = x \in \mathbb{R}, z^{(2)} = y \in \mathbb{R}^2$$
$$f^{(1)}(z) = x^T A_t y, f^{(2)}(z) = -x^T A_t y \tag{136}$$
$$\mathcal{Z} = \mathbb{R} \times \mathbb{R}^2$$

where

$$A_t = \begin{cases} \begin{bmatrix} 1 & -1 \end{bmatrix}, t \text{ is odd} \\ \begin{bmatrix} -1 & 1 \end{bmatrix}, t \text{ is even} \end{cases} \tag{137}$$

The step size $\eta$ is chosen to be $0.005$. The initial points are chosen to be $x_0 = 1, x_{\frac{1}{2}} = 4$ and $y_0 = [2, 3], y_{\frac{1}{2}} = [5, 6]$. The experimental results are presented in Figure 5, where we can see the variant of the OG algorithm makes $r^{tan}(z_t)$ diverge. This result supports the exponentially divergence result in Theorem 5.

## I.6 EXPERIMENTS ON THEOREM 6

We verify Theorem 6 by examples with $k \in \mathbb{N}^*$:

$$z^{(1)} = x \in \mathbb{R}^2, z^{(2)} = y \in \mathbb{R}^2$$
$$f^{(1)}(z) = x^T A_t y, f^{(2)}(z) = -x^T A_t y \tag{138}$$
$$\mathcal{Z} = \mathbb{R}^2 \times \mathbb{R}^2$$

where

$$A_t = \begin{cases} \begin{bmatrix} 2 & -2 \\ -4 & 1 \end{bmatrix}, t = 3k \\ \begin{bmatrix} 1 & 4 \\ 2 & 2 \end{bmatrix}, t = 3k+1 \\ \begin{bmatrix} 2 & 2 \\ 6 & 1 \end{bmatrix}, t = 3k+2 \end{cases} \tag{139}$$

The step size $\eta$ is chosen to be $0.005$. The initial points are chosen to be $z_0 = [4, 5, 2, 3], z_{\frac{1}{2}} = [5, 6, 3, 1]$. The experimental results are presented in Figure 6, where we can see the OG algorithm makes $r_i^{tan}(z_t)$ converge, $i = 0, 1, 2$. This result supports the convergence result in Theorem 6.

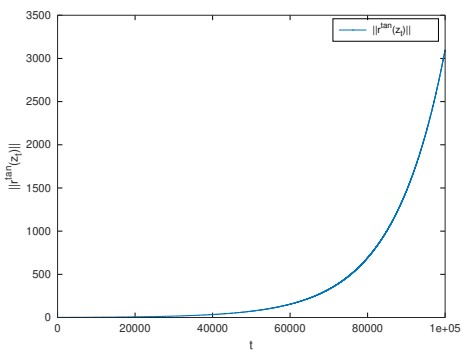

Figure 1: $\|r^{tan}(z_t)\|$ in a periodic game for the RG algorithm

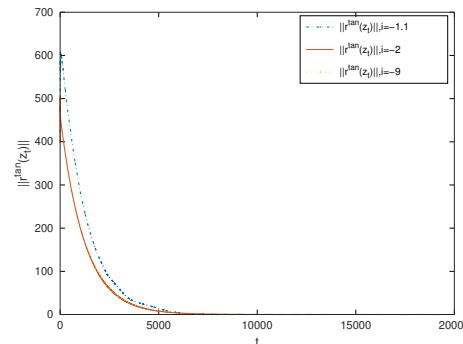

Figure 2: $\|r^{tan}(z_t)\|$ in a convergent perturbed game for the RG algorithm

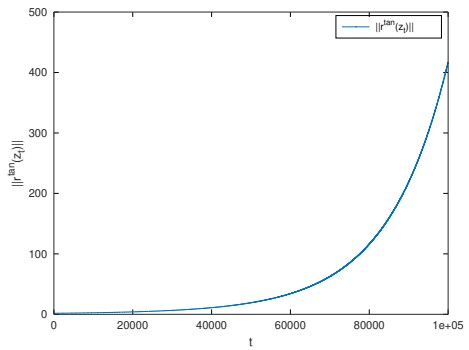

Figure 3: $\|r^{tan}(z_t)\|$ in a periodic game for the ARG algorithm

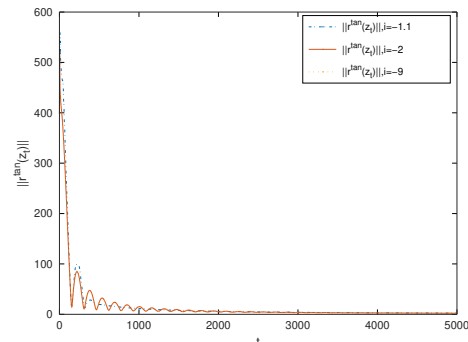

Figure 4: $\|r^{tan}(z_t)\|$ in a convergent perturbed game for the ARG algorithm

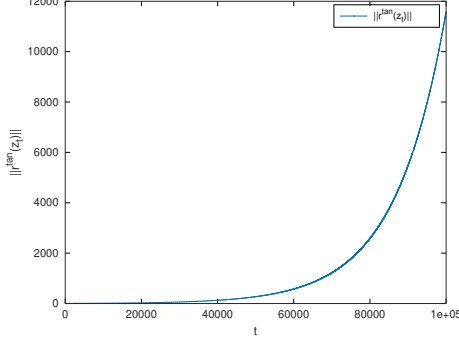

Figure 5: $\|r^{tan}(z_t)\|$ in a periodic game for the modified OG algorithm

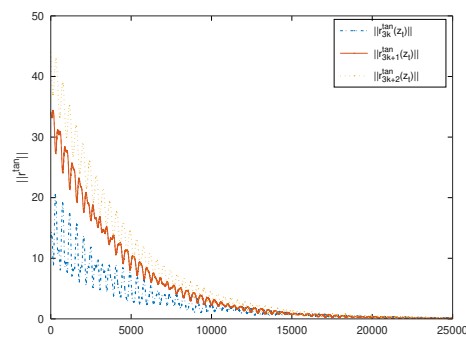

Figure 6: $\|r^{tan}(z_t)\|$ in a periodic game for the OG algorithm

