# OpenReview forum: "Similarity and Separation of Last-Iterate Convergence between Optimism and Reflected Algorithms in Time-Varying Games"
_ICLR.cc/2026/Conference — Submitted to ICLR 2026_

### Official Review · Reviewer_TPem · 2025-10-21

**Soundness:** 3
**Presentation:** 3
**Contribution:** 2
**Rating:** 2
**Confidence:** 4

**Summary:**

This paper studies time-varying games, where the utility function evolves over time. In particular, it considers two cases:
- Fast-convergent perturbed games, in which the total deviation from a fixed game remains bounded, and
- Periodic games, where multiple fixed games repeat in a periodic pattern.
The authors present both positive and negative results for the algorithms RG, ARG, OG, and VOG, as summarized in Table 1.

**Strengths:**

- The paper examines a series of algorithms in this area and presents comprehensive results.
- The presentation is clear, well-organized, and easy to follow.

**Weaknesses:**

My main concern lies in the technical contribution of the results. The paper relies on the BAP assumption to address the convergent perturbed game, but the BAP assumption appear overly strong. In particular, the BAP assumption requires the accumulated deviation to remain constant, which seems unrealistically fast and renders the resulting analysis somewhat trivial.

Since the accumulated deviation is bounded, the result follows directly from the classical analysis of these algorithms in fixed games, with an additional bounded error at each iteration due to the BAP assumption.

**Questions:**

I believe that, to make the results complete, a lower bound on the convergence rate for the perturbed game should be considered. Specifically, let $W_T$ denote the total deviation from the fixed game over $T$ iterations (see [1]). What is the corresponding lower bound on the convergence rate? Furthermore, do the algorithms considered in the paper achieve this lower bound?

I would be inclined to raise my score to accept if the authors could address these questions.

[1] Anagnostides, Ioannis, et al. "On the convergence of no-regret learning dynamics in time-varying games." Advances in Neural Information Processing Systems 36 (2023): 16367-16405.

---

> ### Author Response · Authors · 2025-11-20
>
> Thank you for your insightful review.
>
> Weaknesses:
>
> BAP assumption is not overly strong. For time-varying games, even if the perturbation is converging, proving convergence of the algorithms is non-trivial. Since a converging sequence does not necessarily have a converging partial sum, a weaker assumption may cause difficulties due to the potential divergence resulting from a combination of perturbations. By applying the BAP assumption in our proof, which makes an important addition and significant difference from the time-invariant game, we keep the sequence of perturbation having a converging partial sum to avoid divergence caused by the combination of perturbations. In existing results, (Feng et al, 2023) failed to show that non-BAP perturbation in the PEG algorithm could lead to divergence, but showed that BAP perturbation in the PEG algorithm does lead to divergence although simply convergent perturbation is enough for convergence of the EG algorithm in two-player bilinear games. Besides, (Feng et al, 2023) and (Chen & Yu, 2025) also showed results on convergent perturbed games with BAP assumptions after results on corresponding time-invariant ones are known. In addition, even perturbations converging at a rate as fast as an exponential one exist in reality, like transient responses in analog circuits.
>
> Question:
> I suppose that $W_T$ is related to the regret. We know that this is important, but it is not so related to last-iterate behaviours, which is a more intuitive sight than the regret related to the sum of error from $0$ to $T$. In fact, the extra-gradient algorithm is not a no-regret algorithm at all, but it is still commonly used (see (Cai et al, 2022), (Feng et al, 2023) and (Chen & Yu, 2025), where that algorithm is investigated).

---

> ### Author Response · Authors · 2025-11-23
>
> Dear Reviewer TPem,
>
> We truly appreciate your helpful comments.
>
> We are sending a gentle reminder that the reviewer-author discussion phase is ending. It would be much appreciated if you could kindly tell us the remaining questions, if any. We would kindly ask if you might consider greater support should our responses satisfactorily address your points. We truly appreciate your consideration.
>
> Thank you for your valuable time. May you have a good day.
>
> Best wishes,\
> Authors

---

> > ### Comment · Reviewer_TPem · 2025-11-24
> >
> > ## About BAP
> >
> > > "... but showed that BAP perturbation in the PEG algorithm does lead to divergence ..."
> >
> > Could you please clarify precisely which theorem or result in Feng et al. (2023) establishes this divergence?
> >
> > ## About $W_T$
> >
> > I think the current BAP assumption seems weak (bounded by a constant). I feel like it is possible to achieve last-iterate convergence when $\sum_{t=0}^{T} \max||G_t||\leq O(T^a)$ for some $a\in (0, 1)$. In this case, how should $a$ connect to the last-iterate convergence rate?
> >
> > Conversely, could you show that for any $a>0$, the last-iterate convergence fails?

---

> > > ### Author Response · Authors · 2025-11-24
> > >
> > > Thank you for your comments.
> > >
> > > About BAP: I meant to say Theorem 3.2 in (Feng et al, 2023) and "non-BAP perturbation", not "BAP perturbation". This is a typo.
> > >
> > > About $W_T$: Due to the complex behaviours of $G_t$, although $\sum_{t=0}^{T} \max||G_t||\leq O(T^a)$ for $a\in (0, 1)$ may be enough, there may be significant technical difficulties to prove this. For last-iterate cases, $a>0$ appears to be a new issue, especially for non-strong monotone time-varying problems. We may only discuss the problem in the future.

---

> > > > ### Comment · Reviewer_TPem · 2025-11-26
> > > >
> > > > Thanks for the clarification. I will maintain my score unless the authors can either
> > > >
> > > > - prove divergence for any $a > 0$, or
> > > > - show last-iterate convergence for $a > 0$ and justify it with appropriate lower bounds.

---

> > > > > ### Author Response · Authors · 2025-12-01
> > > > >
> > > > > Thanks for your good comments.
> > > > >
> > > > > For some cases where $\max\\|G_t\\|$ does not exist at all, we have included them in Lemma 2 since $\forall z'\in\mathcal{Z}$, finite $L_{G_t}$ may make $\\|G_t\\|$ be greater than any $N>0$ for a $z\in\mathcal{Z}$ if $\\|z-z'\\|$ is sufficiently great and $\\|G_t(z) − G_t(z')\\|=L_{G_t}\\|z-z'\\|$, and proved that they can be transformed into cases satisfying the BAP assumption. Periodic games can also be interpreted as such games with $a=1$ since their gradient vectors of cost functions are similar to $F_t=F_0+G_t$ where $G_t$ is periodic.

---

### Official Review · Reviewer_7azo · 2025-10-22

**Soundness:** 1
**Presentation:** 2
**Contribution:** 2
**Rating:** 2
**Confidence:** 4

**Summary:**

This paper studies the convergence properties of reflected gradient (RG), accelerated reflected gradient (ARG), and optimistic gradient (OG) in time-varying games including periodic games, and those that convergence fast enough according to a bounded accumulated perturbations assumption (BAP). The paper claims a convergence rate for RG and ARG in games that satisfy a BAP assumption and under bounded action sets but show that they both fail to converge in periodic games. In contrast, the OG method is shown to converge without the BAP assumption and in periodic games so long as the Lipschitz constant is bounded. Empirical simulations are also provided to verify some of the results.

**Strengths:**

The paper provides a somewhat unifying perspective to several algorithms of interest in the context of time-varying games. Including both negative results for RG and ARG as well as strong positive results for the OG method.

**Weaknesses:**

A main weakness of the paper is its technical clarity and rigour as well lack of discussion on some technical details. The proofs are quite difficult to follow even with the proof sketches in the main body. Below I list some concerns, additional issues are listed in the questions section.

**BAP** assumption
The paper uses the BAP assumption to quantify sufficient convergence in perturbed games. However, I've noticed that the assumption includes $\mathcal{Z}$ is bounded meanwhile this is not used in other works such as Feng et al 2023. Furthermore, it is unclear what is meant by $\max \|G_t\|$ is this supposed to mean $\max_{z \in \mathcal{z}}\|G_t(z)\|$? I also cannot see this max in Feng et al 2023 nore in the cited works for the BAP condition. Finally, at several points in the paper the following claim is made:

> Assumption 1 (BAP) implies $\max\|G_t\| = O(\frac{1}{t})$

This is used in line 1206 in Theorem 2 and line 1604 in Theorem 4. I do not see how the BAP summability assumption ($\sum_{t}\max\|G_t\| < \infty$) implies a rate on the last term in the series, in general I think we need more information on the convergence rate.

**BAP** assumption in Section 4.2
On top of the BAP assumption Theorem 4 assume the stronger condition $\sum_{t=0}^\infty t^2\|G_t\|< +\infty$ but there is no discussion why we need this stronger assumption in the ARG case. Intuition or identifying technical challenges on why this case merits a stronger assumption would be helpful to understand the result in the broader context of the paper and existing work.

**Bounded action sets**
All the convergence results (except maybe OG) require bounded action sets yet the divergence negative periodic game results are in the unbounded setting. Discussion on why bounded sets are necessary would be helpful especially when existing works such as Feng et al 2023 do not assume a bounded domain. How would the periodic results change if we were to assume a bounded domain? Additionally, the convergence proofs seem to heavily depend on the bounded domain whereas existing works do not need it, a discussion on this difference and potential limitations of the analysis could be helpful.

**Missing steps and typos** There are quite a few missing steps and typos in the proofs. Please see the question section for more details

**Questions:**

- line 1199 to line 1201 can be expanded with an extra step
- it is unclear where Lemma 6 is invoked in Theorem 2 is it used for line 1209? adding pointers to when lemmas would be helpful
- The transition from equation (51) to equation (52) is not clear
- Where does the inequality in equation (120) come from?
- Weak monotonicity i.e. $\rho$ in Theorems 4-6 are introduced without being discussed in the paper. This assumption/relaxation and its significance  should be included in the background/discussion.
- Where does the VOG algorithm come from (variant of OG)? Is there a reference? In the description of VOG in Theorem 5 seems to suggest that VOG is a proximal like algorithm since $z_{t+1}$ depends on $F_{t+1}$, is this the case?
- Feng et al 2023 shows convergence for EG in the periodic case yet Table 1 seems to suggest that this result is still open. My understanding is that the paper studies both EG and PEG.
- on line 136 it is mentioned that bilinear games are not strictly monotone but in general I  do not think it is true that $\langle F(z_1)-F(z_2), z_1-z_2 \rangle =0$?
- BAP assumption is mentioned on line 141 but only introduced later
- $\mathcal{Z}^{(i)}$ never defined before used in section 2.3
- there seems to be a typo for equations 28 and 27, both are the same?
- In the proof of Lemma 2 why do we have $G_t(z^\star) =0$? A common Nash equilibrium doesn't imply $F_t(z^\star) = F_{\infty}(z^\star)$.

---

> ### Author Response · Authors · 2025-11-20
>
> Thank you for your detailed review.
>
> Weaknesses:
>
> BAP:
>
> For the first issue, if $\\max\\|G_t\\|\\neq O(1/t)$, then there exists $P$ so that $\\sum_{t=0}^T\\max\\|G_t\\|>P\\sum_{t=0}^T1/t>\\infty$.
>
> For the second issue, there is a technical difficulty resulting in the application of stronger assumption than the BAP one: the structure of the ARG algorithm is much more complex than the RG (and PEG and EG) algorithms due to the extra items $\\frac{1}{t+1}(z_0- z_t)-\\frac{1}{t}(z_0 -z_{t-1})$ in Line 4, Algorithm 2 and $\\frac{1}{t+1}(z_0- z_t)$ in Line 5, Algorithm 2, so that it is more difficult to analyze the convergence behaviours of the related partial sum in the ARG algorithm (a converging sequence, which is the case for the perturbation, does not necessarily have a converging partial sum) than in the RG (and PEG and EG) algorithms, which increases the total difficulty in analysis involved in Theorem 4.
>
> Bounded action sets: This is a natural constraint since actual action sets have to be bounded. In the analysis of two-player games, eigenvalues and norms can be applied to prove an exponential convergence rate, but in general cases, the behaviours related to the total sum of $\\|G_t\\|$ are complex. In (Chen & Yu, 2025) which also investigates time-varying games, such a setting has been applied for analyzing the EG and PEG algorithms, so the existing results actually already require this.
>
> Missing steps and typos: See Questions.
>
> Questions:
>
> 1. An extra step will be added in Line 1200: $\\leq\\|F_{T-1}(z_T)-G_{T-1}(z_T)+c_T\\|^2$
>
> 2. Lemma 6 is used in (51). We will add "according to Lemma 6, " before "we have" in Line 1208 above (51).
>
> 3. There is no transition relationship between (51) and (52). However, we will insert "$\\leq P_{t^\*,t\\in[\\lceil\\frac{T}{2}\\rceil, T]}+(\\frac{2}{\\eta}+8L)D_\\mathcal{Z}\\sum_{t=t^\*,t\\in[\\lceil\\frac{T}{2}\\rceil, T]}^{T}\\max\\|G_t\\|+\\frac{2}{\\eta} D_\\mathcal{Z}\\sum_{t=t^\*,t\\in[\\lceil\\frac{T}{2}\\rceil, T]}^{T}\\max\\|G_{t-1}\\|+8\\sum_{t=t^\*,t\\in[\\lceil\\frac{T}{2}\\rceil,, T]}^{T}\\max\\|G_t\\|^2+2\\|F_{T-1}(z_T)\\|\max\\|G_{T-1}\\|+\\max\\|G_{T-1}\\|^2$"before the first "$\\leq$" in (52).
>
> 4. Here we unintentionally used an Assumption (Weak MVI for $E$ with $A=N_\\mathcal{Z}$): For $z^\*\\in\\mathcal{Z}$ such that $0\\in E(z^\*) = F(z^\*) + A(z^\*)$, where $F: \\mathbb{R}^n\\Rightarrow\\mathbb{R}^n$ is single-valued and $A:\\mathbb{R}^n\\Rightarrow\\mathbb{R}^n$ is set-valued maximally monotone, for $E=F+A$ with $\\rho\\leq0$, $\\exists z^\*\\in \\mathrm{Zer}(E)$, $\\langle u,z-z^\*\\rangle\\geq\\rho\\|u\\|^2, \\forall(z,u)\\in\\mathrm{Gra}(E)$.
>
> Note that this was applied to similar problems (Cai et al, 2023; Diakonikolas et al, 2021; Pethick et al, 2022) and mentioned in (Cai et al, 2023) and we will edit properly.
>
> 5. The importance of weak monotonicity has been shown in (Cai et al, 2023). We will add "The following Theorem is inspired by (Cai et al, 2023), which also includes extension on weak monotonicity."
>
> 6. This is an alternative method for reverting Lines 4 and 5 in the OG algorithm since real reverting requires unnatural initial values and we intend to show an alternative structure of the time-varying algorithm. Besides, since $F_t=F_{t+1}$ in time-invariant games, the extension is natural.
>
> 7. Feng et al only investigated two-player bilinear games, which is only one type of cases of monotone multi-player games, so that related results cannot necessarily be extended to all monotone multi-player games.
>
> 8. The proof is as follows:
>
> For the game $\\min_{x\\in\\mathcal{X}}\\max_{y\\in\\mathcal{Y}}x^TAy$ where $z^{(1)}=x$, $z^{(2)}=y$ are actions of the players and $\\mathcal{X}$, $\\mathcal{Y}$ are their corresponding actions sets, the pseudo-gradient $F$ is $\\begin{bmatrix}Ay\\\\-A^Tx\\end{bmatrix}$ so that $\\langle F(z_1)-F(z_2), z_1-z_2\\rangle=\\begin{bmatrix}A(y_1-y_2)\\\\-A^T(x_1-x_2)\\end{bmatrix}^T\\\\\\begin{bmatrix}x_1-x_2\\\\y_1-y_2\\end{bmatrix}=(y_1-y_2)^TA^T(x_1-x_2)-(x_1-x_2)^TA(y_1-y_2)=0$.
>
> 9. It is not proper to define the BAP assumption right in the middle of the introduction. However, we will insert "(defined in Assumption 1, Section 2.2, which means the time-varying game converges fast enough)'' after "under BAP assumptions" in the introduction.
>
> 10. We will insert ``where $\\mathcal{Z}^{(i)}$ means the action set of Player $i$'' before ". By the general definition" in Section 2.3.
>
> 11. They are not the same. There is a difference between $z_{t-1}$ and $z_{t+1}$.
>
> 12. We should have included $G_t(z^\*)=0$ in Lemma 2. This is a typo.
>
> References:
>
> J. Diakonikolas, C. Daskalakis, and M. Jordan. Efficient methods for structured nonconvex-nonconcave min-max optimization. International Conference on Artificial Intelligence and Statistics, 2021.
>
> T. Pethick, P. Latafat, P. Patrinos, O. Fercoq, and V. Cevherå.
> Escaping limit cycles: Global convergence for constrained nonconvex-nonconcave minimax problems. In ICLR 2022.

---

> ### Author Response · Authors · 2025-11-23
>
> Dear Reviewer 7azo,
>
> We truly appreciate your constructive comments helping us improve this article.
>
> We are sending a gentle reminder that the deadline for the reviewer-author discussion phase is rapidly approaching, and your problems mentioned in "Weaknesses" and "Questions" have already been addressed properly in our comments and the revised paper. We would greatly appreciate it if you could take some time to review our response. If there are any additional questions or clarifications, please do not hesitate to let us know. We would also appreciate it if you could consider providing greater support, should our responses satisfactorily address the points you raised. We truly appreciate your consideration and look forward to your further comments. May you have a good day.
>
> Best wishes,\
> Authors

---

### Official Review · Reviewer_8ohC · 2025-11-01

**Soundness:** 3
**Presentation:** 3
**Contribution:** 3
**Rating:** 6
**Confidence:** 3

**Summary:**

This paper investigates the convergence properties of several first-order algorithms, specifically Reflected Gradient (RG), Accelerated Reflected Gradient (ARG), and Optimistic Gradient (OG), in the context of multi-player, time-varying monotone games. The authors analyze two distinct time-varying settings: (1) periodic games, where the game's cost functions repeat with a fixed period, and (2) fast convergent perturbed games, where the game converges to a limit $F_{\infty}$ and the perturbation satisfies a Bounded Accumulated Perturbations (BAP) assumption.

The paper's core contribution is a "separation" of behaviors between these algorithms:

- Convergent Perturbed (BAP) Case: The RG and ARG algorithms are shown to be robust to this type of perturbation. The paper proves last-iterate convergence rates of $O(1/\sqrt{T})$ for RG (Theorem 2) and an optimal $O(1/T)$ for ARG (Theorem 4), assuming the perturbation converges sufficiently fast.
- Periodic Case: In sharp contrast, the paper provides counterexamples showing that both RG (Theorem 1) and ARG (Theorem 3) can diverge exponentially in periodic games.
- In addition, the standard OG algorithm is proven to be robust in both settings, achieving a best-iterate convergence rate of $O(1/\sqrt{T})$ in time-varying games with a common Nash equilibrium, which includes the periodic case (Theorem 6). However, a slight variant of OG (VOG) is shown to fail.

**Strengths:**

- One strength of this paper is the separation it identifies, which is the exponential divergence of RG and ARG in periodic games (Theorems 1 and 3).
- The paper also shows several positive results for the convergent perturbed (BAP) case, including a $O(1/\sqrt{T})$ last-iterate rate for RG (Theorem 2) and an optimal $O(1/T)$ last-iterate rate for ARG (Theorem 4) in this time-varying setting.
- In addition, the authors also show that the robustness of the standard OG algorithm to periodic games (Theorem 6). This contrasts sharply with the failure of PEG/OGDA in periodic games (as cited from Feng et al. (2023)) and the failure of the VOG variant shown in this paper (Theorem 5).

**Weaknesses:**

- The positive results for the OG algorithm in periodic games (Theorem 6) and the analysis for the BAP case (Lemma 2) rely on the assumption of a common Nash equilibrium $z^*$ for all games $F_t$. This assumption seems to be strong and counter-intuitive for a time-varying game. The paper motivates periodic games by citing ``seasonal changes or market competitions'', but in such scenarios, one would expect the equilibrium itself to be periodic, not static. This assumption severely limits the practical implications and generality of the paper's key positive result (Theorem 6).
- In addition, the optimal $O(1/T)$ rate for ARG (Theorem 4) requires the perturbation $G_t$ to converge very rapidly, specifically $\sum_{t=0}^{\infty}t^{2}||G_{t}||<\infty$. This is even stronger than the standard BAP assumption (Assumption 1, which is $\sum_{t=0}^{\infty} \max ||G_t|| < \infty$). While the paper shows the rate degrades to $O(1/\sqrt{T})$ under the weaker Assumption 1, the $O(1/T)$ optimal rate relies on this impractical, convergence condition.
- The "variant of OG" (VOG) is introduced as a foil to the standard OG algorithm. However, its definition and motivation are confusing. Theorem 5 states it is created by modifying Line 5 of Algorithm 3 to use $F_{t+1}$ instead of $F_t$. Algorithm 3 (OG) uses $F_t$ to compute both $z_{t+1/2}$ (Line 4) and $z_{t+1}$ (Line 5). The modification $z_{t+1} = z_{t+1/2} + \eta F_{t+1}(z_{t-1/2}) - \eta F_{t+1}(z_{t+1/2})$ (based on and Algorithm 3) appears to require oracle access to the gradient function $F_{t+1}$ at time $t$. It is not clear to me why this is a "natural variant" to study, as it seems to break the assumptions of online learning.
- While the paper is technically dense, the reliance on proof sketches and deferrals to the appendix for all details makes the core technical arguments difficult to fully assess. It would be better to include explanations/intuitions on the proofs of Theorem 4/5/6 in the main text.

**Questions:**

Besides the question in the Weakness section:
- Could the authors please clarify the definition and motivation for the VOG algorithm (Theorem 5)? The use of $F_{t+1}$ in the update rule seems to require an oracle for the next time step's gradient function.
- The $O(1/T)$ rate for ARG requires $\sum t^2 ||G_t|| < \infty$. Is this assumption necessary, or is it an artifact of the proof technique (e.g., the potential function $V_t$ in Eq. 70)? Is there any evidence to suggest that the $O(1/T)$ rate is impossible under the standard BAP assumption (Assumption 1)?

---

> ### Author Response · Authors · 2025-11-20
>
> Thank you for your insightful review.
>
> Weakness:
>
> 1. There are significant technical difficulties for such analysis in general time-varying cases. More importantly, the common Nash equilibrium is not highly restrictive since two-player bilinear games are quite extensively discussed, like (Feng et al, 2023) and (Feng et al, 2024).
>
> 2. This is indeed an artifact of proof technique. However, the convergence condition is not impractical since even exponentially convergent disturbance is realistic, e.g., transient responses in analog electronic circuits, and we aim at showing a comparable convergence result to time-invariant games. Such an assumption is applied due to technical difficulties introduced by extra items in the ARG algorithm which makes the structure of the ARG algorithm be much more complex than the RG (and PEG and EG) algorithms due to the extra items $\frac{1}{t+1}(z_0- z_t)-\frac{1}{t}(z_0 -z_{t-1})$ in Line 4, Algorithm 2 and $\frac{1}{t+1}(z_0- z_t)$ in Line 5, Algorithm 2 so that it is more difficult to analyze the convergence behaviours of the related partial sum in the ARG algorithm (a converging sequence, which is the case for the perturbation, does not necessarily have a converging partial sum) than in the RG (and PEG and EG) algorithms, which increases the total difficulty in analysis involved in Theorem 4.
>
> 3. This is an alternative method for reverting Lines 4 and 5 in the OG algorithm since real reverting appears unnatural and we intend to show an alternative structure of the time-varying algorithm. Besides, since $F_t=F_{t+1}$  in time-invariant games, the extension is natural.
>
> 4. The proof sketch of Theorem 6 is not included for the simplicity of the proof. The other two sketches of the proofs are already included. It was not possible to extend them for the page limits.
>
> Questions:
>
> 1. See Weakness 3.
>
> 2. See Weakness 2.

---

> > ### Comment · Reviewer_8ohC · 2025-11-27
> >
> > Thanks for the detailed response from the authors. However, I am still confused about the notations in the modification of OG. First, I do not understand how Line 4 and Line 5 in OG can be reverted since Line 4 is computing $z_{t+1/2}$ that is used in Line 5. Second, while $F_t=F_{t+1}$ in time-invariant games, since now the game is changing over time, it is more standard to only have access to the information till round $t$ when we compute $z_{t+1}$.

---

> > > ### Author Response · Authors · 2025-11-29
> > >
> > > Thanks for your new comments.
> > >
> > > With the intention of reverting Lines 4 and 5, we mean to calculate $z_{t}$ and $z_{t+1/2}$ (which is an intermediate step after $t$) at the same time, instead of calculating $z_{t+1}$ and $z_{t+1/2}$ (which is an intermediate step before $t+1$) at the same time. At the time $t=0$, if we replace $F_t$ in Line 4 with $F_{t-1}$ instead of replacing $F_t$ in Line 5 with $F_{t+1}$, we can indeed avoid using $F_{t+1}$ in time $t$, but then we must have $F_{-1}$ at time $0$, which is even more unnatural, since we can prepare initial actions before $t=0$, but having $F_t$ before the game even starts is weird. In addition, (Feng et al, 2023) also involve the gradient vector $F_{t+1}$ at time $t$, although that is in a different algorithm (called alternating negative momentum method).

---

> ### Author Response · Authors · 2025-11-23
>
> Dear Reviewer 8ohC,
>
> Thanks to your insightful and helpful comments.
>
> We are sending a gentle reminder that the deadline for the reviewer-author discussion phase is rapidly approaching, and your problems in "Weaknesses" and "Questions" have been properly addressed in our comments. We would greatly appreciate it if you would take some time to examine our response. If you have any further questions or require further clarification, please do not hesitate to let us know.
>
> May you have a good day.
>
> Best wishes,\
> Authors

---

### Official Review · Reviewer_i3wz · 2025-11-01

**Soundness:** 4
**Presentation:** 4
**Contribution:** 3
**Rating:** 6
**Confidence:** 3

**Summary:**

This paper investigates the Reflected Gradient, Accelerated Reflected Gradient, and Optimistic Gradient algorithms in L-smooth time-varying monotone games. It establishes a surprising separation: while RG and ARG achieve strong convergence rates in fast-converging perturbed games, they both diverge exponentially in periodic games. In sharp contrast, the standard OG algorithm is robust and converges (O (1/\sqrt{T})) in periodic games, while a minor variant of it diverges.

**Strengths:**

1. Theoretical: The core finding of the paper is the "surprising" separation in convergence behaviors. It demonstrates that while standard OG converges in periodic time-varying games, the closely related RG, ARG, and even a slight variant of OG (VOG) all diverge exponentially. This is a novel and important insight into the specific properties that enable convergence in non-stationary environments.
 2. Empirical: The theoretical claims are illustrated and verified by Section I, that directly test the divergence and convergence predictions of all six theorems.
 3. Clarity: The paper is exceptionally well-written. It clearly motivates the problem — less computational cost, defines the gap it aims to fill, and summarizes its findings effectively in the introduction and in Table 1.

**Weaknesses:**

I have the following main concerns:
 1. Weak Motivation for the VOG Variant: The paper introduces VOG algorithm primarily to show it diverges, highlighting the specificity of the standard OG's success. However, the paper provides little motivation for why this specific variant is a "natural" one to consider. The finding is further diluted by the proof's demonstration that this VOG is equivalent to the RG algorithm in bilinear games, which was already shown to diverge (Theorem 1).
 2. Strong Assumption for OG Robustness: The celebrated robustness of the standard OG algorithm (Theorem 6) is proven only under the assumption that the sequence of time-varying games shares a common Nash equilibrium. This is a significant limitation, as many realistic periodic or time-varying games would likely feature an equilibrium that also varies with time.

I might miss some key points, please correct me if anything has been misunderstood.

**Questions:**

1. Could you elaborate on the choice of the VOG algorithm (Theorem 5)?

2. It is proved a best-iterate rate for OG (Theorem 6) and list last-iterate convergence as future work. Do you have any intuition as to whether standard OG achieves last-iterate convergence in the periodic setting?

3. The robustness of the standard OG algorithm (Theorem 6) is shown under the strong assumption of a common Nash equilibrium across all time-varying games. This seems quite restrictive for general periodic or time-varying scenarios. Could the convergence guarantee of OG be extended to the more general setting?

---

> ### Author Response · Authors · 2025-11-20
>
> Thank you for your insightful review.
>
> Weakness 1 and Question 1:
> This is an alternative method for reverting Lines 4 and 5 in the OG algorithm since real reverting appears unnatural and we intend to show an alternative structure of the time-varying algorithm. Besides, since $F_t=F_{t+1}$  in time-invariant games, the extension is natural.
>
> Weakness 2 and Question 3: The common Nash equilibrium is not highly restrictive since two-player bilinear games are quite extensively discussed, like (Feng et al, 2023) and (Feng et al, 2024). For the possibility of extending the result on OG to the more general setting, we believe that this is possible, but there might be significant technical difficulties. The reason that we apply the BAP assumption is the following technical difficulty: for time-varying games, the boundedness of actions at different times is not trivial, and since a converging sequence does not necessarily have a converging partial sum, by applying the BAP assumption, we keep the sequence of perturbation having a converging partial sum to avoid divergence caused by the combination of perturbation. The BAP assumption also involves the proof of Lemma 7 in our paper. Note that if the perturbation is simply converging, proving a convergence of the algorithms is not trivial. (Feng et al, 2023) showed that non-BAP converging perturbation in the EG algorithm could lead to divergence but did not show that non-BAP converging perturbation in the PEG algorithm could lead to convergence.
>
> Question 2:
> Probably yes, considering the similar results in RG and ARG.

---

> ### Author Response · Authors · 2025-11-23
>
> Dear Reviewer i3wz,
>
> Thanks to your insightful and helpful comments.
>
> We are sending a gentle reminder that the deadline for the reviewer-author discussion phase is rapidly approaching, and your problems in "Weaknesses" and "Questions" have already been properly addressed in our rebuttal. We would greatly appreciate it if you would take some time to examine our response. If you would like to raise any additional questions or require additional clarification, please do not hesitate to let us know.
>
> May you have a good day.
>
> Best wishes,\
> Authors

---

### Meta-Review · Area_Chair_8cQR · 2025-12-25

**Summary:**

This paper studies time-varying games in which the utility function evolves over time. It considers two cases: the first is fast-convergent perturbed games, where the total deviation from a fixed game remains bounded (under the BAP assumption); the second is periodic games, where multiple fixed games repeat in a periodic pattern. The authors present both positive and negative results for four algorithms, i.e., RG, ARG, OG, and VOG, with findings summarized in Table 1 of their paper.

While the initiative is good, a major concern shared by Reviewer 7azo and Reviewer TPem lies on the BAP assumption made in this work, i.e.,
$$
G_t := \nabla g_t, Z \text{ is bounded, and } \sum_{t=0}^{\infty} \max \| G_t \| < \infty.
$$
Specifically, Reviewer TPem thinks that this assumption is strong and feels that the results in this paper could be obtained by straightforward modifications of existing analyses for fixed games. Moreover, Reviewer 8ohC points out that Theorem 5 requires the perturbation $G_t$ to converge very rapidly, i.e.,
$\sum_{t=0}^{\infty} t^2 \| G_t \| < \infty $ which is an even stronger requirement than the BAP assumption. The authors did make some efforts in clarifying the concerns on the BAP assumption. However, after reviewing the authors' reply, Reviewer TPem further raises a fundamental question, which is whether $\sum_{t=0}^T \max \| G_t \| = O(T^a), \text{ where } a \in (0,1)$, is sufficient for convergence. The reviewer mentioned that they would raise the score if the authors could argue that for any $a \in (0,1)$, the dynamics will lead to divergence (to justify that the BAP assumption may not be strong at all).

Both Reviewer 8ohC and 7azo also noted that the proofs are hard to parse. Reviewer 7azo, in particular, provides a list of steps that require further elaboration.

I encourage the authors to strengthen their work by addressing these lingering issues. It would also be beneficial if the authors could illustrate a couple of concrete real-world examples where such time-varying games become relevant. In its final version, this work will make a solid contribution to learning and games.

**Reviewer Concerns:**

Reviewer 7azo asked several clarification questions, and the authors have addressed all of them. For the lingering concerns, please see the "Summary" section above.

**Reviewer Scores:**

Given that some of the concerns of Reviewer 7azo and Reviewer TPem listed above are still there, it is unlikely that they will upgrade their scores, in my opinion. Reviewer 8ohC also mentioned that, after reading the authors' rebuttal, they still find it difficult to understand the modification of OG, e.g., "First, I do not understand how Line 4 and Line 5 in OG can be reverted since Line 4...". Hence, it might be elusive for them to fully support the paper.

---

### Decision · Program_Chairs · 2026-01-26

Reject